# Design and Research of Series Actuator Structure and Control System Based on Lower Limb Exoskeleton Rehabilitation Robot

Chenglong Zhao [1,2,3], Zhen Liu [2,*], Liucun Zhu [3,4,*] and Yuefei Wang [1]

1 School of Mechanical and Marine Engineering, Beibu Gulf University, Qinzhou 535011, China; zhaochenglong@bbgu.edu.cn (C.Z.); yfwang@bbgu.edu.cn (Y.W.)
2 Department of Integrated Systems Engineering, Nagasaki Institute of Applied Science, Nagasaki 851-0193, Japan
3 Advanced Science and Technology Research Institute, Beibu Gulf University, Qinzhou 535001, China
4 Research Institute for Integrated Science, Kanagawa University, Yokohama 259-1293, Japan
* Correspondence: z-liu_zhen@nias.ac.jp (Z.L.); l-lczhu@bbgu.edu.cn (L.Z.)

**Abstract:** Lower limb exoskeleton rehabilitation robots have become an important direction for development in today's society. These robots can provide support and power to assist patients in walking and movement. In order to achieve better interaction between humans and machines and achieve the goal of flexible driving, this paper addresses the shortcomings of traditional elastic actuators and designs a series elastic–damping actuator (SEDA). The SEDA combines elastic and damping components in parallel, and the feasibility of the design and material selection is demonstrated through finite element static analysis. By modeling the dynamics of the SEDA, using the Bode plot and Nyquist plot, open-loop and closed-loop frequency domain comparisons and analyses were carried out, respectively, to verify the effect of damping coefficients on the stability of the system, and the stiffness coefficient $k_s$ = 25.48 N/mm was selected as the elastic element and the damping coefficient $c_s$ = 1 Ns/mm was selected as the damping element. A particle swarm optimization (PSO)-based algorithm was proposed to introduce the fuzzy controller into the PID control system, and five parameters, namely the the fuzzy controller's fuzzy factor ($k_e$, $k_{ec}$) and de-fuzzy factor ($k_{p1}$, $k_{i1}$, $k_{d1}$), are taken as the object of the algorithm optimization to obtain the optimal fuzzy controller parameters of $k_e$ = 0.8, $k_{ec}$ = 0.2, $k_{p1}$ = 0.5, $k_{i1}$ = 8, $k_{d1}$ = −0.1. The joint torque output with and without external interference is simulated, and the simulation model is established in the MATLAB/Simulink environment The results show that when fuzzy PID control is used, the amount of overshooting in the system is 14.6%, and the regulation time is 0.66 s. This has the following advantages: small overshooting amount, short rise time, fast response speed, short regulation time, good stability performance, and strong anti-interference ability. The SEDA design structure and control method breaks through limitations of the traditional series elastic actuator (SEA) such as its lack of flexibility and stability, which is very helpful to improve the output effect of flexible joints.

**Keywords:** series elastic–damping actuator; lower limb exoskeleton; rehabilitation robot; fuzzy control





## 1. Introduction

The concept of rehabilitation robots first emerged in the 20th century; however, limited technological advancements at that time hindered the application of robots in the field of rehabilitation. In recent years, significant progress has been achieved in rehabilitation robotics due to continuous advancements in mechanical engineering control technology and sensor technology. This progress spans various medical domains, including neural rehabilitation, physical rehabilitation, and musculoskeletal rehabilitation [1]. These robots can assist patients in walking and movement, aiding in joint mobility, muscle exercises, and balance training. They hold particular importance for individuals with lower limb impairments and amputees, contributing to accelerated rehabilitation processes, and improved rehabilitation experiences and outcomes, ultimately enhancing patients' quality of life.

The key factors determining the performance of lower limb exoskeleton rehabilitation robots lie in their mobility, specifically how they demonstrate natural, smooth, and adaptable human-like walking during motion. This aspect is highly crucial for the practicality of the robots and their interaction with humans [2]. There are various ways to achieve fluidity in robotic walking, such as designing appropriate gait generation algorithms, adjusting gait parameters through control algorithms to suit changing environments and tasks, or equipping the robot with a variety of sensors, such as inertial measurement units (IMUs), force sensors, visual sensors, etc., to acquire real-time environmental and robot status information. This feedback can be utilized to modify gait and posture, ensuring the robot's motion is stable. Furthermore, the incorporation of flexible materials and structures in robots allows impact to be reduced during collisions, facilitating smoother and more fluid motion.

A serial elastic actuator (SEA) is a driving system commonly used in robotics, exoskeleton devices, and other mechanical systems. Its main principle involves utilizing elastic elements such as springs, cables, or gas bags to store and release energy, thereby facilitating the driving and motion of mechanical systems. The distinguishing feature of this actuator is its capacity to store and release energy, which helps mitigate the impact and vibrations that traditional rigid actuators might generate [2]. This enhances the mobility of robots, elevates the comfort and naturalness of exoskeleton devices, and promotes smoother and more efficient motion. The concept and principle design of serial elastic actuator (SEAs) were introduced by the MIT Leg Laboratory [3], and they have been applied to bipedal robots, enabling torque control [4]. Researchers, led by Marco Hutter at ETH Zurich, developed the high-performance quadruped robot StarlETH, incorporating linear SEAs as the drivers for each joint [5]. These joint SEAs not only enhance the accuracy of torque control loops but also ensure energy efficiency, allowing StarlETH to exhibit remarkable mobility in unstructured environments. Venema et al. employed cable-driven joint SEAs, translating the force from linear elastic elements into joint torsion. They simplified the motor to an ideal velocity source model, validating the output force characteristics of SEAs [6].

SEAs have the advantages of structural shock resistance, low mechanical output impedance, and strong ability to adapt to the environment compared with rigid actuators [7], but in the simplification of the kinetic model of the elastic element, the kinetic model is imperfect as a result of simplifying the SEA into a pure stiffness link or damping link, and it is difficult to realize flexible driving like in human skeletal muscle by relying on the elastic element only. In addition, in the analysis of the dynamics model, many scholars regard the motor as an ideal velocity output source or position output source, ignoring the inertia characteristics of the motor itself, which causes large errors [8]. In addition, most drive control methods do not take into account the existence of various external disturbances in the system itself, while the perturbation problem is unavoidable in practical application [9].

In view of the above problems, this paper firstly improves the SEA, and proposes a structure combining the spring and damper. The damper is used to improve the damping device in order to attenuate the vibration produced by the impact very quickly [10], and the damper is combined with the elastic element to constitute a new type of actuator, the series elastic–damping actuator (SEDA). Solidworks is used to complete the three-dimensional model design of SEDA, and finite element static analysis of the key components is used to verify the reasonableness and reliability of the structure and material selection. Secondly, the motor is regarded as an ideal force output source, and since the motor output force is directly proportional to the excitation current (voltage) in the range of output capacity [11] and the change time of the excitation current is almost negligible for a mechanical system such as an SEDA, it is more reasonable to simplify the motor as a kind of force output source. Furthermore, the open-loop and closed-loop system transfer functions of the force source drive are obtained through the Laplace transform. The influences of the elasticity coefficient and damping coefficient on the system are analyzed. Then, the system's stability is analyzed using the Nyquist stability criterion, and its force output bandwidth and output

impedance are obtained by the Bode plot, which proves the reasonableness of this structure. Finally, for the control of the output force of the SEDA, the fuzzy control is introduced on the basis of traditional PID control algorithms and is carried out on the MATLAB/Simulink platform. The Simulink platform was simulated and analyzed to verify the correctness of the fuzzy PID controller design.

## 2. Background

In our previous research efforts, we designed an exoskeleton robot for lower limb rehabilitation. This robot includes three degrees of freedom at the hip, knee, and ankle joints. It is equipped with RMD-X8pro1:9V3 servo motors and utilizes CAN communication. The three-dimensional model of the robot is shown in Figure 1. The designed structure involves direct rigid joint actuation, known as rigid control, providing rapid response speed and high control precision. The exoskeleton features a length adjustment mechanism to accommodate patients of different heights. The entire robot is constructed using 7075 aluminum material, meeting the anticipated strength requirements. However, the total weight of the robot has reached 35 kg, rendering it relatively cumbersome. Participant fitting results are depicted in Figure 2, illustrating that the robot falls short of the lightweight design objective. Moreover, the rigid mechanical structure is uncomfortable for users and could potentially cause secondary harm. Previous research also revealed that significant impact forces are generated when the human heel contacts the ground. Hence, the design of lower limb exoskeleton rehabilitation robots should ensure structural strength and precise positioning while also enhancing human–environment interaction, all while taking into account the lessons learned from our previous studies.

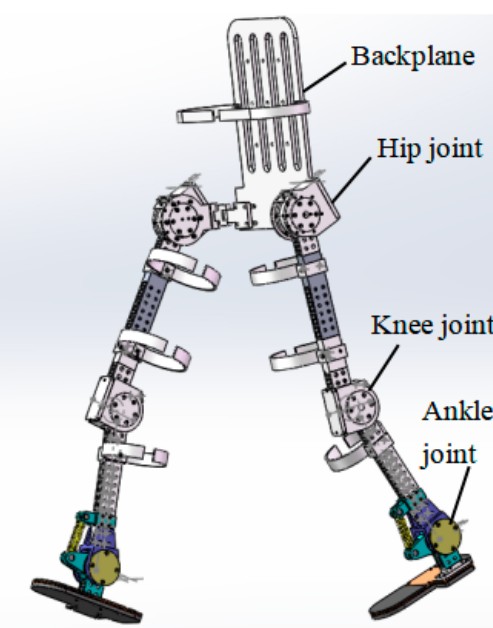

**Figure 1.** Three-dimensional model of lower limb exoskeleton rehabilitation robot.

The experiments were conducted using Quanser's Optitrack optical 3D motion capture system equipped with six FLEX 3 infrared cameras for spatial 3D localization, together with a 3D force measuring table and Motive data analysis software version 2.0.0. A three-dimensional space with a length of 5 m, a width of 4.4 m and a height of 2.6 m was installed indoors, and one healthy male subject was selected to participate in the lower limb joint motion capture experiment (age 25 years old, body weight of 75 Kg, height of 175 cm), taking walking as the basic motion mode, taking the horizontal road surface as the basic constraints, and combining the characteristics of the muscle activity groups in the human body's walking with the passive infrared optics' reflecting principle. The lower limbs of the human body were positioned in a non-linear, non-contrasting, non-directional, and

non-directional direction. A total of 17 marker points were pasted on the lower limbs in a non-linear and asymmetric way, walking barefoot, and the lower limbs were synchronized with the three-dimensional force measuring table, with a sampling frequency of 100 Hz, so that the motion parameters of the human body's various joints and the GRF (ground support return force) curves could be obtained, as shown in Figure 3.

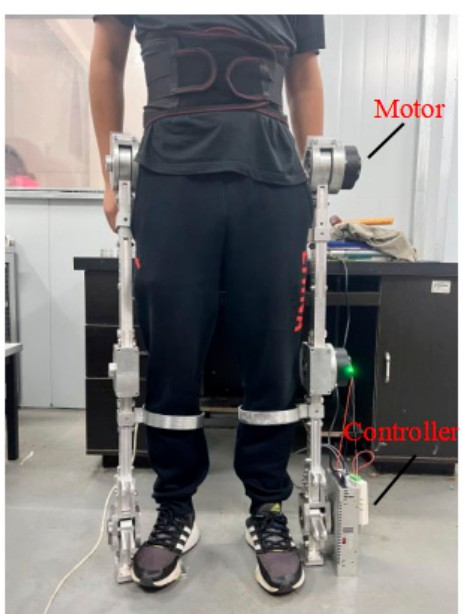

**Figure 2.** Trial fitting effect image.

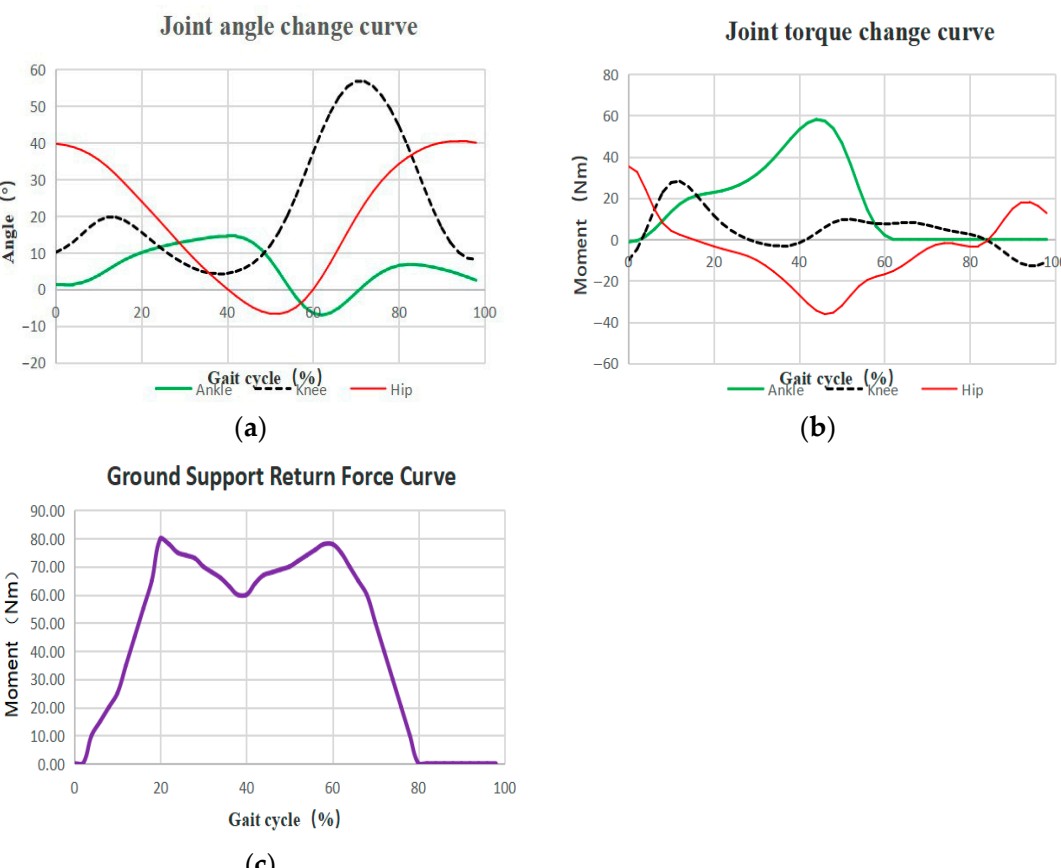

**Figure 3.** Joint motion parameters and GRF curves: (**a**) joint angle change; (**b**) joint output torque; (**c**) barefoot walking GRF curve.

In Figure 3b, it can be found that the average moment generated by the hip and knee joints is larger in the normal walking process of a person, so we take these two joints as the active joints and install the external joint drive system. Through the GRF curve in Figure 3c, it can be seen that there are two peaks appearing in the curve, the first one is the impact peak, also called the passive peak, which comes from the force recoiled by the ground to the foot and calf at the initial moment when the heel initially touches the ground, and the second peak, also called the active peak, which occurs at about the middle moment, which comes from the force generated by the foot to support the weight of the body. According to the angle changes in each joint in Figure 3a, angle limits are set at the hip and knee joints, as shown in Figure 4. A passive joint is used at the ankle joint, as shown in Figure 5, which consists of a calf and foot assembly attached to the ankle joint. A spring is used to generate an auxiliary torque during dorsiflexion, featuring an adjustable initial ankle angle and magnitude of the dorsiflexion auxiliary torque, which collects energy at a certain gait stage and is able to release the stored energy at an appropriate time.

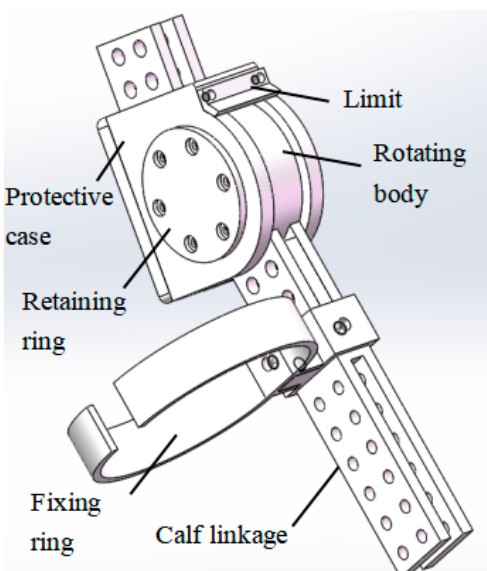

**Figure 4.** Joint angle limitation.

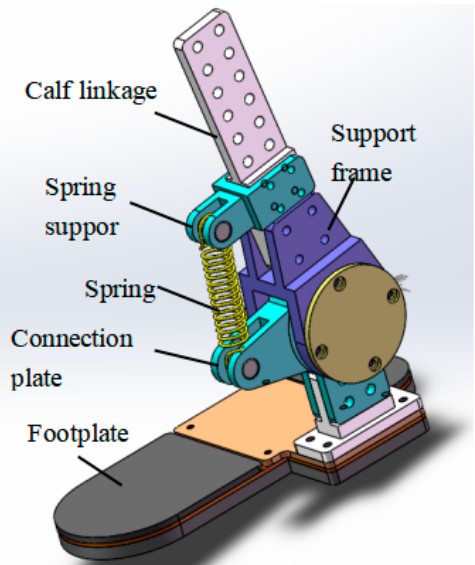

**Figure 5.** Ankle joint structure.

The design for enhancing the flexibility of lower limb exoskeleton rehabilitation robots should not be limited solely to the ankle joint, but should encompass a holistic approach to the robot's integration, taking into account the hip and knee joints. Consequently, a series elastic execution mechanism is proposed to be combined with rigid structures, applied to various joints of the robot. This approach, while preserving the existing functionality, simplifies the robot's structure, reduces its weight, ensures safe interaction with patients, and enables the implementation of innovative rehabilitation strategies.

## 3. SEDA (Series Elastic Damping Actuator) Design

### 3.1. Overall Structure of SEDA

The overall structure of the SEDA designed in this study is illustrated in Figure 6. It primarily consists of a servo motor, ball screw, linear springs, spring plates, guide rods, dampers, bearings, and other components. The driver incorporates eight linear springs mounted on guide rods, forming the foundation of a traditional SEA. Additionally, four dampers are added to create the SEDA. The distribution of elastic elements is symmetric, ensuring stable operation of the driver. The entire driver is partitioned by spring plates, achieving flexibility in both forward and reverse directions.

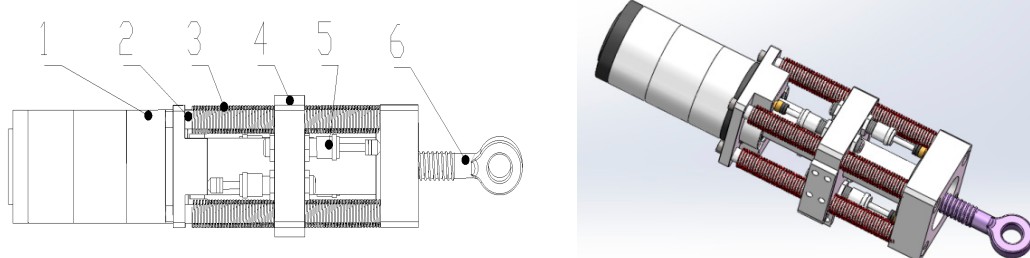

**Figure 6.** Overall structure of SEDA: 1—servo motor; 2—guiding shaft; 3—mold spring; 4—spring separator; 5—damper; 6—ball screw.

The operational principle of SEDA is as follows: In SEDA, the motor is driven by a control signal, causing the transmission shaft to rotate. The rotational movement of the transmission shaft drives the nut to compress the spring without an oil liner, extending or retracting the ball screw, and subsequently propelling the crank structure to rotate. The crank structure imparts rotational motion to the thigh or shin. When the thigh or shin encounters external impact, the entire SEDA, apart from the spring partitioning components, moves collectively along the guide axis. This transmits the applied force to the dampers and springs. The springs themselves lack damping capability, and without dampers, bouncing issues might arise in the leg. The combination of springs and dampers not only absorbs the impact of external load fluctuations but also attenuates vibrations, achieving stability and realizing mechanical flexibility and stability in the joints. The improved effect is depicted in Figure 7.

### 3.2. Finite Element Static Analysis

During walking, impacts are generated on the SEDA, and its structure is susceptible to potential damage. The material selection for SEDA needs to balance lightweight design with ensuring appropriate strength and deformation characteristics. Therefore, finite element analysis (FEA) is performed using the SolidWorks Simulation plugin to create displacement contour maps, stress contour maps, deformation contour maps, and more [12]. This aids in obtaining a better understanding of the SEDA's structural behavior. Given the relatively slow walking speed of the lower limb exoskeleton rehabilitation robot, static analysis is focused on key components.

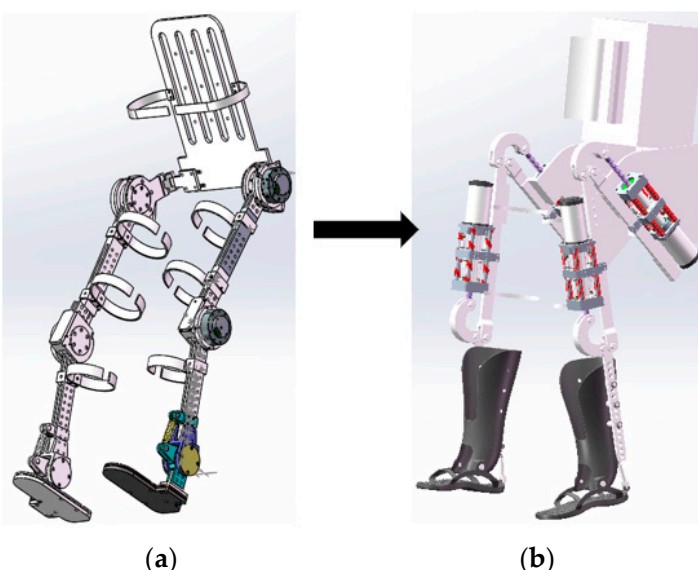

|  |  |
|:---:|:---:|
| (**a**) | (**b**) |

**Figure 7.** Assembly diagram of SEDA before and after improvement: (**a**) original assembly diagram; (**b**) improved assembly diagram.

### 3.2.1. SEDA Subjected to Axial Thrust

The SEDA subjected to radial thrust is depicted in Figure 8. Since components such as the lead screw and bearings are standard parts, their strength analysis is currently omitted. Apart from standard components, all other materials employ 7050 aluminum alloy with a yield strength of 435 MPa and a tensile strength of 495 MPa.

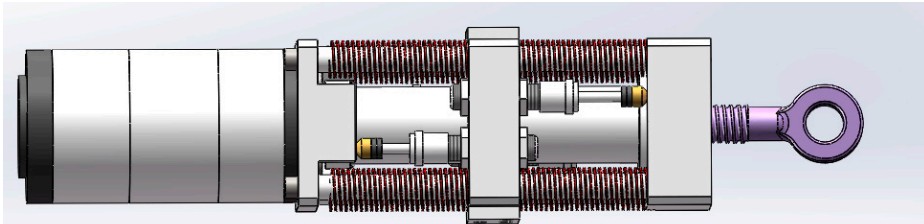

**Figure 8.** Axial thrust model of SEDA.

As shown in Figure 9, with the increasing radial force, the critical region for potential SEDA failure is the fixed sleeve. To simplify the mechanical structure, elements such as bolts, springs, and motors that have little relevance to the strength analysis were removed for the strength verification. The mesh type used is a solid mesh, the Jacobian point of the high quality mesh is 16 points, the cell type is a solid cell, the cell size is 3.54919 mm, the tolerance is 0.17746 mm, the total number of cells is 10,098, and the total number of nodes is 18,472. As illustrated in Figure 9a, the simplified geometric model is depicted, while Figure 9b represents the meshed model. The resulting responses are displayed in Figure 9c,d. The green arrow in the figure indicates the direction of fixture fixation and the purple arrow indicates the direction of applied force. From the figures, it is apparent that when a radial force F of 5000 N is applied, the maximum equivalent stress that may lead to failure is 45.7 MPa, and the maximum overall deformation is 0.046 mm. Both values remain within acceptable limits.

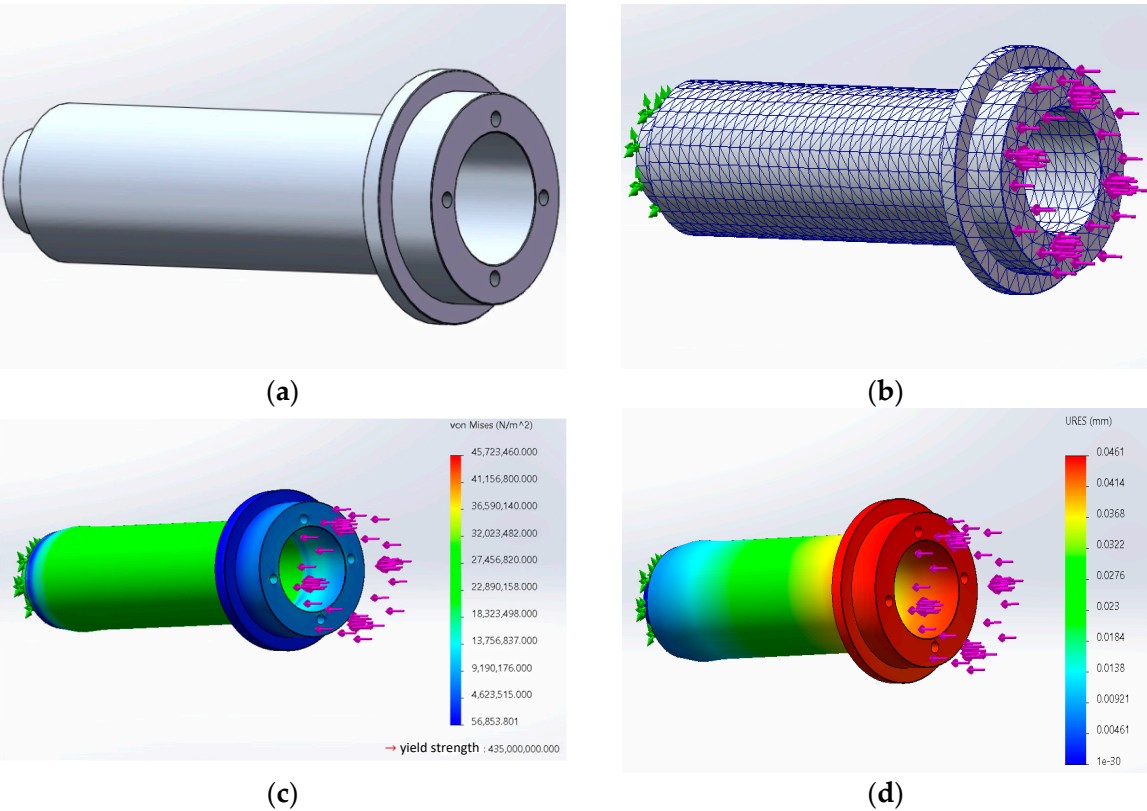

**Figure 9.** Finite element analysis of SEDA structure part I: (**a**) model diagram; (**b**) mesh diagram; (**c**) deformation diagram; (**d**) stress diagram.

3.2.2. SEDA Subjected to Radial Thrust

The model of a SEDA subjected to axial thrust is shown in Figure 10. As the screw shaft and bearings are standard components, their strength analysis is temporarily omitted. Apart from standard components, the remaining materials utilize a 7050 aluminum alloy with a yield strength of 435 MPa and a tensile strength of 495 MPa.

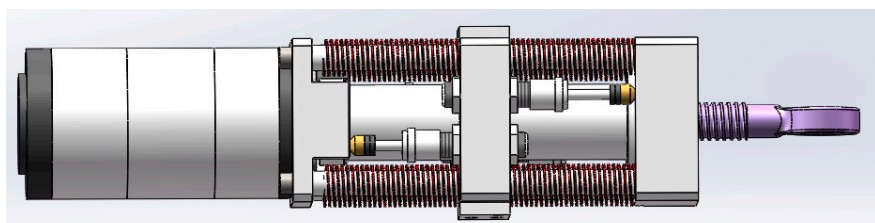

**Figure 10.** SEDA radial thrust model.

As shown in Figure 11, with the continuous increase in axial force, the susceptible locations for potential damage in the SEDA are the guide rod and the drive plate. The mechanical structure is simplified to some extent by excluding structural components such as bolts, springs, and motors that have minimal relevance to the strength analysis. The mesh type used is a hybrid mesh, the Jacobian point of the high quality mesh is 16 points, the cell type is a shell cell, the cell size is 5.94583 mm, the tolerance is 0.297291 mm, the total number of cells is 41,237, and the total number of nodes is 74,032. As illustrated in Figure 11a, a simplified geometric model is presented, while Figure 11b represents the corresponding mesh model. The results of the analysis are displayed in Figure 11c,d. The purple arrow in the figure indicates the direction of the applied force.From the figures, it is evident that when a radial force of 3000 N is applied, the maximum equivalent stress prone

to damage reaches 396 MPa, and the maximum overall deformation is 0.5 mm; both values fall within the permissible range.

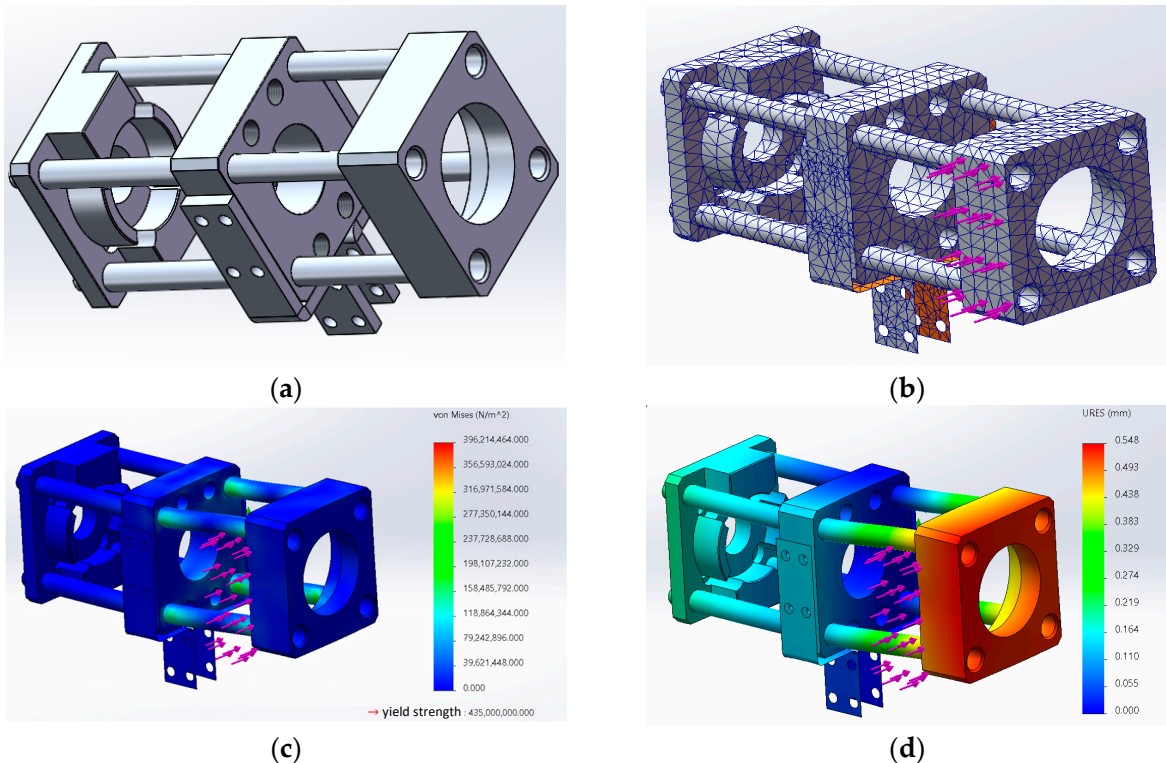

**Figure 11.** SEDA structural finite element analysis II: (**a**) model diagram; (**b**) mesh diagram; (**c**) deformation diagram; (**d**) stress diagram.

Based on the above finite element strength analyses of key components of the SEDA under radial or axial loads, it can be concluded that the structural design and material selection of SEDA are rational.

## 4. Analysis of the Dynamic Characteristics of SEDA

During the walking process, the impact force generated upon foot–ground contact is transmitted through the lower limbs to various joints in the body. Rigid driving mechanisms can result in instantaneous vibrations that significantly affect the stability of walking. The SEA involves installing a set of elastic elements between the power source and the load, allowing for better adaptation to external environments and simulating the movement characteristics of human muscles. This approach meets the requirements of rehabilitation robots for smoothness, adaptability, and safety [13]. Currently, elastic elements, primarily springs such as linear and torsional springs, are commonly used. However, relying solely on these elements only approximates the properties of human muscles and falls short of achieving flexible driving akin to human skeletal muscles. Dampers are devices designed to enhance damping by quickly attenuating vibrations caused by impacts. By paralleling dampers with elastic elements, a spring–damping system is formed, generating a new driving system model. This constitutes a significant part of the primary research focus of this paper.

### 4.1. SEDA Design Principle

Common actuation methods for drivers include motor-driven, hydraulic-driven, and pneumatic-driven systems, among which motor-driven systems stand out due to their simplicity in structure, high transmission efficiency, and precise positioning. Therefore, a servo motor is chosen as the power source in this context, with a lead screw as the

transmission mechanism and a spring as the elastic element, ultimately linked to the load. The traditional schematic design principle of the elastic-driven actuator is illustrated in Figure 12.

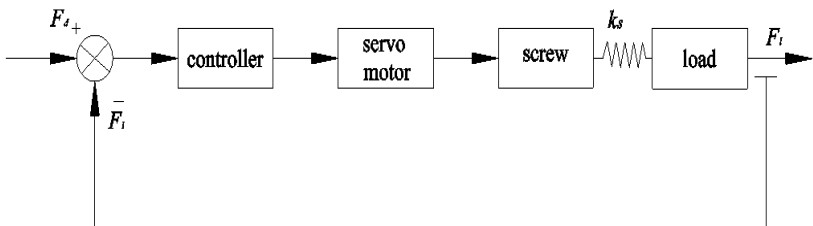

**Figure 12.** Schematic diagram of elastic actuator design.

Building upon the foundation of traditional elastic-driven systems, the incorporation of dampers serves the primary purposes of shock absorption and energy dissipation. By absorbing and dissipating vibrational energy, the dampers reduce the amplitude and duration of system oscillations. When the system experiences external disturbances or excitations, the dampers suppress and control the vibrations. Damping forces introduce a damping effect on system oscillations, diminishing the amplitude and quickly returning the system to its equilibrium position. The schematic diagram of an elastic-driven system with damping is depicted in Figure 13. In the context of the SEDA (series elastic–damping actuator) model, the desired output force is represented as $F_d$, and the feedback force is denoted as $F_l$. The motor's rotation drives the movement of the lead screw, which, in turn, operates the system's output through the elastic link.

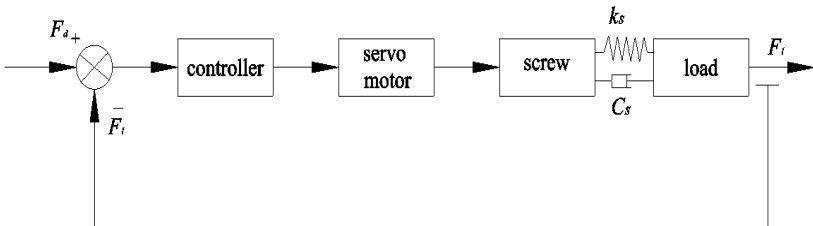

**Figure 13.** Schematic diagram of series elastic−damping actuator design.

Considering the application context of the SEDA, the choice of control method for the power source involves viewing the driving motor as an ideal force output source. In comparison to position/velocity source control methods, this approach offers higher precision. However, it requires accounting for factors such as the damping of the drive system, resulting in a more complex structure [14]. By simplifying the SEDA schematic, a dynamic model based on force source control is derived, as illustrated in Figure 14. To achieve a stable output force and reduce computational complexity, a PID control algorithm is employed in this study. This algorithm calculates the control output based on the error between the current state and the desired state of the system, striking a balance between stability, accuracy, and response speed [15].

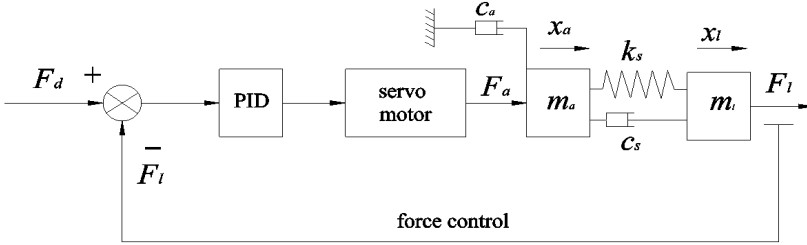

**Figure 14.** SEDA dynamic model based on force source control.

In the aforementioned model, Fa represents the propulsive force generated as the motor's output torque is transmitted through the oil-free, bushing-driven lead screw, resulting in push displacement $x_a$ along the axial direction. The displacement $x_l$ is the displacement generated at the drive output that propels the load, i.e., the human joint. In the context of this equation, $k_s$ denotes the stiffness coefficient of the spring, $c_s$ stands for the damping coefficient of the damper, $c_a$ represents the damping coefficient of the motor, $m_a$ pertains to the mass of the lead screw transmission part, and ml corresponds to the mass of the output-end load. The rotational motion of the motor generates torque, and the relationship between the thrust $F_a$, which is the force output through lead screw transmission, and the actual output force $F_l$ is as follows:

$$F_a - F_l = m_a \ddot{x}_a - c_a \dot{x}_a \tag{1}$$

The force $F_l$ resulting from the interaction of the spring and the damper, as well as the differential equation for the force exerted on the lead screw, is given by:

$$F_l = k_s(x_a - x_l) + c_s(\dot{x}_a - \dot{x}_l) \tag{2}$$

After undergoing Laplace transformation, the open-loop transfer function is obtained as follows:

$$F_l(s) = \frac{(k_s + c_s s)F_a - sk_s(c_s + m_a s)X_l(s)}{k_s + (c_s + c_a)s + m_a s^2} \tag{3}$$

### 4.2. Analysis of Undamped Characteristics in SEDA Open-Loop System
### 4.2.1. SEDA Output Bandwidth

When the robot interacts with the environment, it can be assumed that the load end of the actuator is fixed, i.e., $X_l = 0$. From the above formula, the open-loop bandwidth transfer function between the actual output force $F_l$ and the desired output force $F_a$ can be derived as follows:

$$G_1(s) = \frac{F_l(s)}{F_a(s)} = \frac{k_s + c_s s}{m_a s^2 + (c_s + c_a)s + k_s} \tag{4}$$

When the damping coefficient $c_s = 0$, the equation becomes the following:

$$G_2(s) = \frac{F_l(s)}{F_a(s)} = \frac{k_s}{m_a s^2 + c_a s + k_s} \tag{5}$$

In the robot walking process, the load mass of 100 kg is the limit, and the external load force is about 1000 N. Referring to ISO 10243 (international standard) to select the parameters of the spring [16], the total length of the selected spring is 65 mm, the inner diameter is 8mm, and the outer diameter is 16mm. Finally, the stiffness and compression ratio of the five types of springs, namely, small light load, light load, medium load, heavy load, and super heavy load, are determined as shown in Table 1.

**Table 1.** Spring parameters.

| Rigidity (N/mm) | Maximum Load (N) | Compression Rate | Outer Diameter (mm) | Inner Diameter (mm) | Length (mm) |
|---|---|---|---|---|---|
| 6.37 | 205 | 50% | 16 | 8 | 65 |
| 13.24 | 343 | 40% | 16 | 8 | 65 |
| 24.03 | 500 | 32% | 16 | 8 | 65 |
| 48.35 | 755 | 24% | 16 | 8 | 65 |
| 75.41 | 980 | 20% | 16 | 8 | 65 |

The motor and lead screw have a mass of $m_a = 2$ kg, and the motor damping coefficient is $c_a = 0.2$ Ns/mm. When the damper is not installed ($c_s = 0$), the eight springs are divided into two groups by spring separators, with each group containing four springs. The stiffness coefficients $k_s$ for these groups are 25.48 N/mm, 52.96 N/mm, 96.12 N/mm, 193.4 N/mm,

and 301.64 N/mm. The Nyquist and Bode plots for the open-loop transfer function of this system are illustrated in Figure 15.

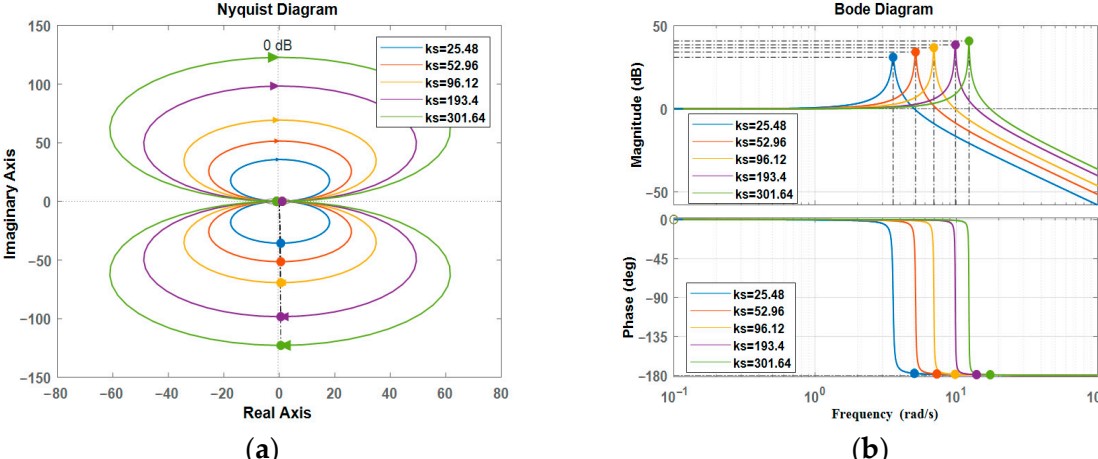

**Figure 15.** Open-loop system bandwidth graph: (**a**) Nyquist diagram; (**b**) Bode diagram.

From Figure 15, it can be observed in (a) the Nyquist plot that all characteristic roots of the open-loop transfer function are negative, and the Nyquist plot does not pass through the point $(-1, j0)$. This implies that the corresponding closed-loop system is stable. However, as the stiffness coefficient $k_s$ increases continuously, the Nyquist curve gradually approaches the point $(-1, j0)$, indicating a weakening of system stability. Simultaneously, in (b) the Bode plot, it is evident that the system exhibits good amplitude and phase tracking characteristics in the low-frequency range. The phase curve does not intersect with $-180°$, signifying excellent stability. Nevertheless, as the $k_s$ coefficient increases, the system's stability deteriorates, leading to phase lag in the high-frequency range and causing overshoot in the system response [17].

### 4.2.2. SEDA Output Impedance

In the scenario of free movement at the load end, while maintaining the desired output force $F_d = 0$, the relationship between the force exerted at the load end $F_l$ and the displacement $X_l$ at the load end is expressed as follows:

$$Z_1(s) = \frac{F_l(s)}{X_l(s)} = \frac{-sk_s(c_s + m_a s)}{k_s + (c_s + c_a)s + m_a s^2} \tag{6}$$

When the damping coefficient $c_s = 0$ in the equation:

$$Z_2(s) = \frac{F_l(s)}{X_l(s)} = \frac{-k_s m_a s^2}{k_s + c_a s + m_a s^2} \tag{7}$$

The output impedance of a mechanical system is a significant parameter that describes the relationship between system output and external loads or the environment. Represented as $Z(s)$, the output impedance of the SEDA is often nonlinear and can potentially change with variations in frequency or vibration modes. Keeping all parameters constant, the impedance Bode plot and Nyquist plot of the system are plotted for different stiffness coefficients, as shown in Figure 9.

From Figure 16a,b, it can be seen that, it is evident that the output impedance $Z(s)$ is relatively small in the low-frequency range. However, in the high-frequency range, as the spring stiffness coefficient $k_s$ increases, the impedance value of the system becomes progressively larger. This indicates a trend toward a more rigid connection between the driver and the load, resulting in decreased system stability and rendering it less suitable for use as a driver.

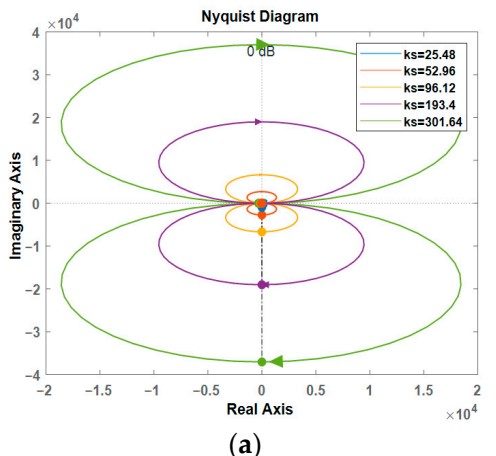
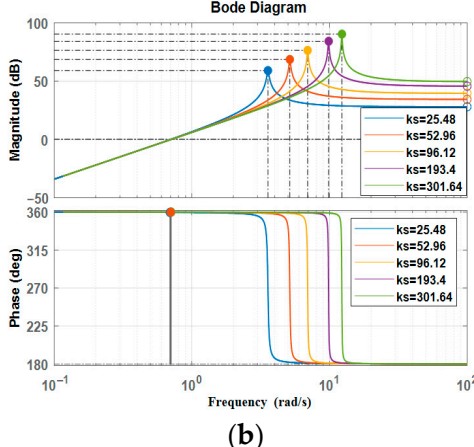

(**a**)  (**b**)

**Figure 16.** Open-loop system output impedance plot: (**a**) Nyquist diagram; (**b**) Bode diagram.

### 4.2.3. SEDA Impact Resistance

In the walking process of lower limb exoskeleton rehabilitation robots, when the feet make contact with the ground, a significant impact is exerted on the robot's body, which can even cause damage to its components. Furthermore, the impact force between the robot and the ground may potentially lead to secondary harm to the patients. Therefore, effectively reducing the generation of impact force holds crucial research significance.

During the walking process, when the feet make momentary contact with the ground, the duration is short, making it difficult to achieve adjustments solely through the expansion or contraction of the lead screw. The SEDA can effectively utilize elastic elements and dampers to control the impact force within a manageable range. The velocity at the moment of contact is defined as $V_l$, and the force exerted at the load end is the actual output force $F_l$. Consequently, the impact power $P_l$ experienced by the robot is:

$$P_l = F_l V_l \tag{8}$$

Given the aforementioned model, it is known that the mass of the rigid body, which is the load, is $m_l$. The initial velocity of the rigid body when it makes contact with SEDA is $v_0$, and the compression of the elastic element is $x_l$. The stiffness coefficient of the spring is $k_s$, and the damping coefficient of the damper is $c_s$. According to Newton's second law, the resulting motion differential equation can be derived as follows:

$$\frac{d\left(m_l \dot{x}_l\right)}{dt} = k_s x_l + c_s \dot{x}_l \tag{9}$$

At time $t = 0$, the moment when the load $m_l$ collides with the SEDA, the initial conditions are $x_l(0) = 0$ and $\dot{x}_l(0) = v_0$. This yields:

$$x_l = \frac{v_0}{\omega_l \sqrt{\xi^2 + 1}} \sin \sqrt{\xi^2 + 1} \, \omega_l t \tag{10}$$

where $\omega_l = \sqrt{k_s / m_l}$ and $\xi = \frac{c_s}{2\sqrt{k_s m_l}}$ are the natural frequency and damping ratio of the system, respectively.

Differentiating the above formula, we obtain the velocity of the drive end as $v_l$:

$$v_l = \dot{x}_l = v_0 \cos \sqrt{\xi^2 + 1} \, \omega_l t \tag{11}$$

After performing the Laplace transform, the equation becomes:

$$V_l(s) = v_0 \frac{s}{s^2 + (\xi^2 + 1)\omega_l^2} \tag{12}$$

When the feet make contact with the ground, the load experiences the spring force and damping force $F_l$:

$$F_l = k_s x_l + c_s \dot{x}_l \tag{13}$$

Taking into account the above formulas, we arrive at:

$$P_l = v_0^2 \left( \frac{k_l}{2\omega_l \sqrt{\xi^2 + 1}} \sin 2\sqrt{\xi^2 + 1}\omega_l t + \frac{c_s}{2} \cos 2\sqrt{\xi^2 + 1}\omega_l t + \frac{c_s}{2} \right) \tag{14}$$

After performing the Laplace transform, the equation becomes:

$$P_l(s) = \frac{v_0^2}{2} \frac{c_s s^2 + c_s s + 2k_s + 4c_s(\xi^2 + 1)\omega_l^2}{s^2 + 4(\xi^2 + 1)\omega_l^2} \tag{15}$$

When there is no damping component, i.e., $c_s = 0$, and the damping ratio is $\xi = 0$, the formula becomes:

$$P_l(s) = v_0^2 \frac{k_s}{s^2 + 4\omega_l^2} \tag{16}$$

From the above formula, it can be observed that in the absence of damping, the shock resistance of the SEDA is mainly influenced by the stiffness coefficient of the spring and the system's natural frequency. With a constant rigid body mass ml and initial velocity $v_0$, the primary factor affecting shock resistance is the spring's stiffness coefficient $k_s$. The resulting bode diagram of the system is shown in Figure 17a, while the simulation yields the GRF curve of the lower limb exoskeleton rehabilitation robot, as shown in Figure 17b.

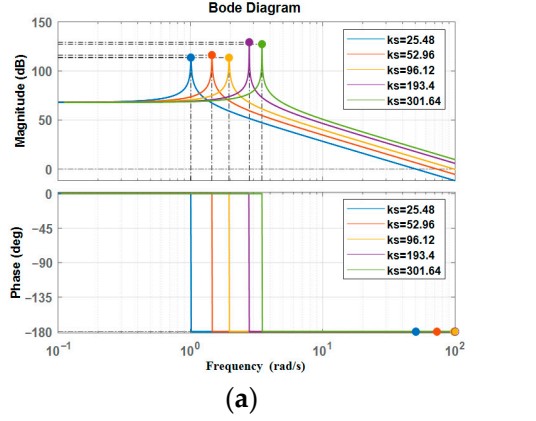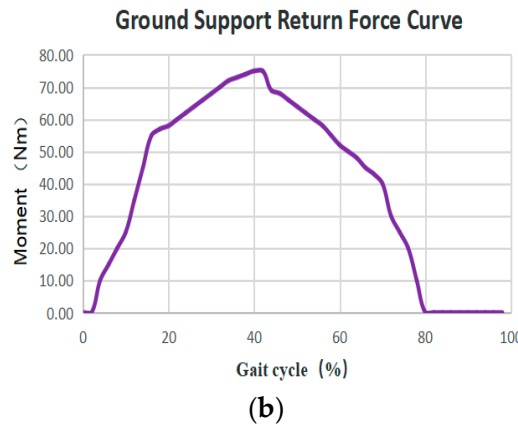

(**a**)　　　　　　　　　　　　　　　(**b**)

**Figure 17.** System shock resistance diagram: (**a**) system Bode diagram; (**b**) robot GRF curve.

As can be seen from Figure 17, the system impact resistance is relatively stable in the low-frequency band, but in the high-frequency band, as the spring stiffness coefficient $k_s$ increases, the impact power $P_l(s)$ of the system becomes bigger and bigger, and the smaller the stiffness coefficient is, the impact power of the system decreases more and more obviously in the high-frequency band. Compared to the barefoot walking GRF plot, the peak shock value is missing, the slope of the line segment in the initial stage has become flat, and the vertical shock rate has decreased, which is mainly due to the elastic and damping elements in the SEDA eliminating part of the shock force and cushioning the vibration.

### 4.3. Analysis of Damped Characteristics in the SEDA Open-Loop System

Referring to the elastic element stiffness coefficient of $k_s$ = 25.48 N/mm, dampers are added to this structure while ensuring an effective load capacity of no less than 100 kg. The damping coefficient $c_s$ of the damper is chosen as follows: 0 Ns/mm, 0.1 Ns/mm, 0.5 Ns/mm, 1 Ns/mm, 3 Ns/mm, and 5 Ns/mm. Other parameters remain unchanged. The output bandwidth, output impedance, and impact resistance Bode plots of the open-loop transfer function of the system are shown in Figure 18.

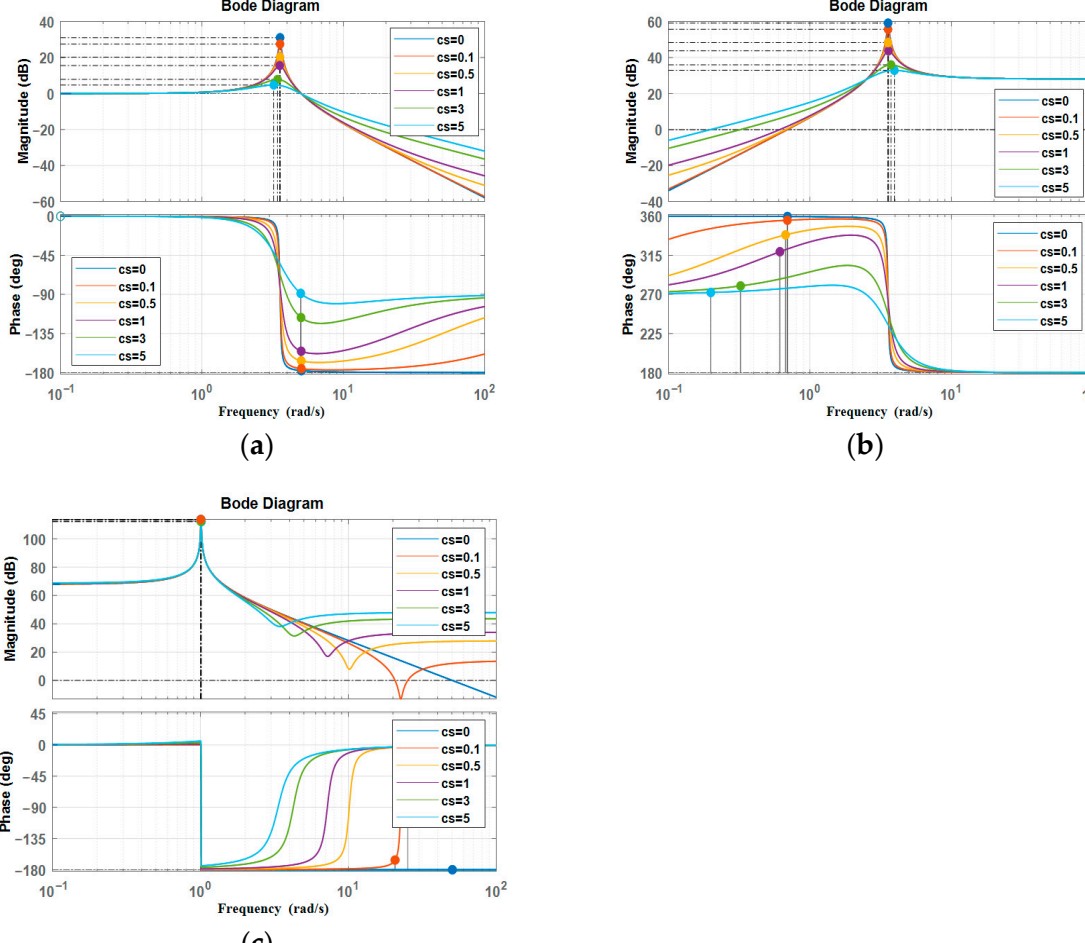

**Figure 18.** Bode diagram of open-loop system's damping characteristics analysis: (**a**) output bandwidth Bode diagram; (**b**) output impedance Bode diagram; (**c**) shock absorption capability Bode diagram.

From the Bode plots shown in Figure 18a–c, which respectively represent the output bandwidth with damping, the output impedance with damping, and the impact resistance, it can be observed that the system exhibits good amplitude tracking and phase tracking characteristics in the low-frequency range. The phase curve does not intersect −180°, indicating the system's stability. As the damping coefficient $c_s$ increases, the resonance peak becomes smaller, indicating improved system stability. However, a higher damping coefficient leads to reduced comfort for individuals wearing the exoskeleton.

### 4.4. Analysis of SEDA Closed-Loop System Characteristics

To enhance system stability and accuracy, a closed-loop control feedback strategy is employed. This involves measuring the system's output and comparing it to the desired output. Subsequently, the controller output is adjusted based on the error signal, aiming to bring the actual output closer to the desired output.

Based on the aforementioned analysis, we have derived the output force $F_a$ of the SEDA. By substituting the formula into the actual output force $F_l$ and introducing PID control, we obtain:

$$F_a = \left(k_p + \frac{k_i}{s} + k_d s\right)(F_d - F_l) \tag{17}$$

As a result, the closed-loop feedback function of the system is obtained:

$$F_l(s) = \frac{(A_3 s^3 + A_2 s^2 + A_1 s + A_0)F_d(s) - s^2 k_s (c_s + m_a s)X_l(s)}{B_3 s^3 + B_2 s^2 + B_1 s + B_0} \tag{18}$$

In the above equation:

$$\begin{cases} A_0 = k_i k_s \\ A_1 = k_i c_s + k_p k_s \\ A_2 = k_p c_s + k_d k_s \\ A_3 = k_d c_s \end{cases} \quad \begin{cases} B_0 = k_i k_s \\ B_1 = k_i c_s + k_p k_s + k_s \\ B_2 = k_p c_s + k_d k_s + c_s + c_a \\ B_3 = k_d c_s + m_a \end{cases}$$

Under the closed-loop control, with the load end fixed, the closed-loop transfer function of the SEDA's drive force output characteristics can be obtained from the above equation:

$$G_3(s) = \frac{F_l(s)}{F_d(s)} = \frac{A_3 s^3 + A_2 s^2 + A_1 s + A_0}{B_3 s^3 + B_2 s^2 + B_1 s + B_0} \tag{19}$$

Under closed-loop control, the relationship between the output force $F_l$ at the load end and the displacement $X_l$ of the load motion is given by:

$$Z_3(s) = \frac{F_l(s)}{X_l(s)} = \frac{-s^2 k_s (c_s + m_a s)}{B_3 s^3 + B_2 s^2 + B_1 s + B_0} \tag{20}$$

### 4.4.1. Simulation of Output Torque in Closed-Loop SEDA System

Based on the established dynamic model and derived closed-loop transfer function, a closed-loop control system based on PID control was implemented using Matlab/Simulink. The output torque characteristics were analyzed for different spring stiffness ($k_s$) and damper coefficient ($c_s$) values. After tuning, the parameters were set as $k_p$ = 8, $k_i$ = 15, $k_d$ = 0.5. Figure 19 shows the control system diagrams and output torque characteristic curves for different spring stiffness values while keeping $c_s$ = 1 Ns/mm. Figure 20 illustrates the control system diagrams and output torque characteristic curves for different damper coefficient values with $k_s$ = 25.48 N/mm.

The simulation curves reveal that when a step input of 5 Nm is applied, the overshoot of the joint output torque increases with the increase in spring stiffness and decreases with the reduction in the damper coefficient. This implies that a higher spring stiffness leads to quicker system response time but greater fluctuations in output torque. On the other hand, an increase in damper coefficient results in reduced system response time and decreased fluctuations in output torque. Therefore, considering the trade-off between system bandwidth, output impedance, shock resistance, and output torque overshoot, the final choice is to employ a relatively smaller spring stiffness of $k_s$ = 25.48 N/mm as the elastic component and a damper coefficient of $c_s$ = 1 Ns/mm as the damping element.

### 4.4.2. Stability Analysis of SEDA Closed-Loop System

Taking $k_p$ = 8, $k_i$ = 15, $k_d$ = 0.5, and keeping other parameters consistent with the open-loop system, with a damping coefficient $c_s$ = 1 Ns/mm and a spring stiffness coefficient $k_s$ = 25.48 N/mm, a comprehensive analysis of the closed-loop system's characteristics was conducted. The results are presented in Figure 21, where Figure 21a,b provide a comparative analysis of the closed-loop and open-loop systems' output bandwidth through

Bode and Nyquist plots. Figure 21c,d present a comparative analysis of the closed-loop and open-loop systems' output impedance through Bode and Nyquist plots.

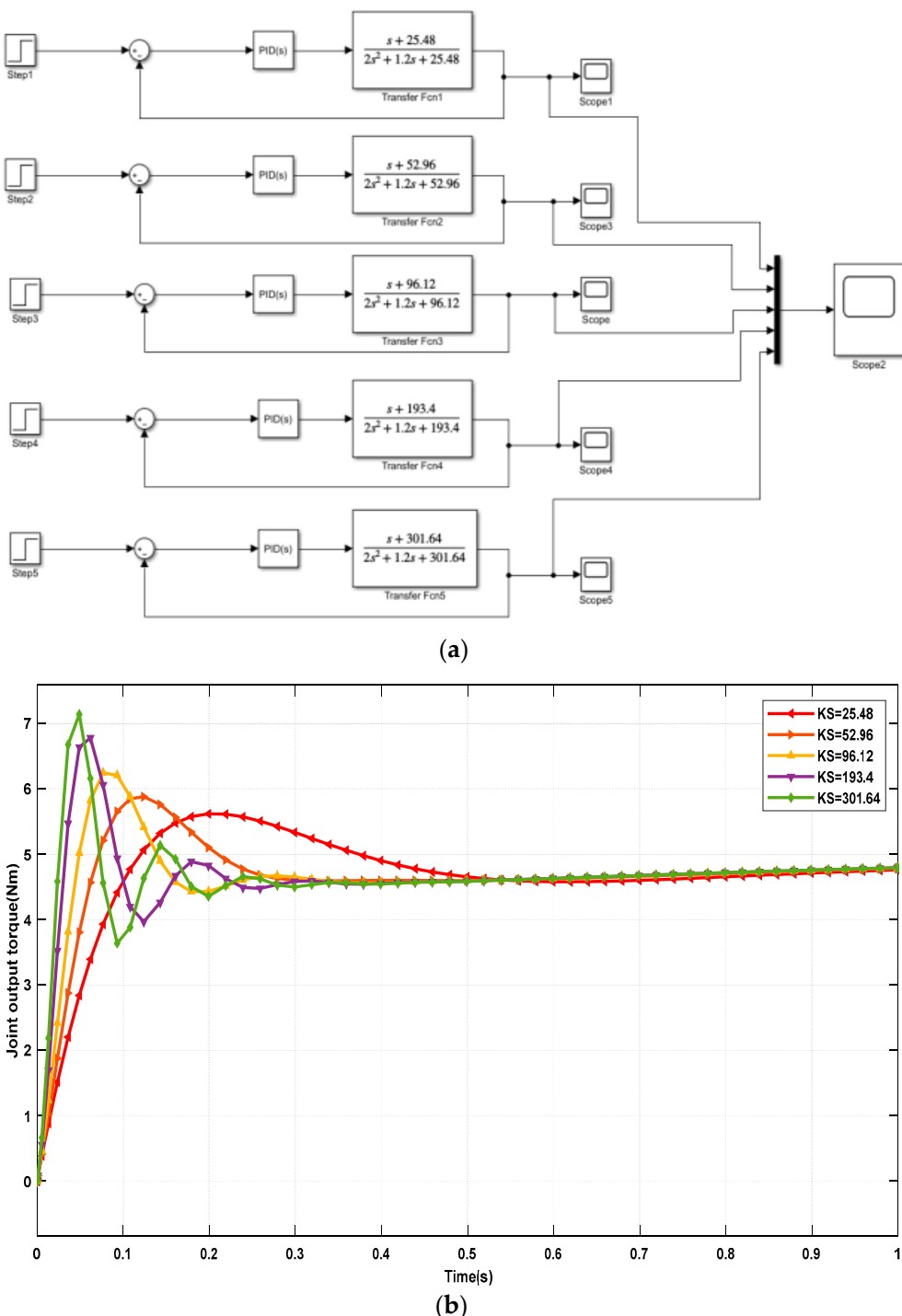

(**a**)

(**b**)

**Figure 19.** Time-varying curves of output torque for different stiffness coefficients: (**a**) Simulink control system; (**b**) output torque variation curve.

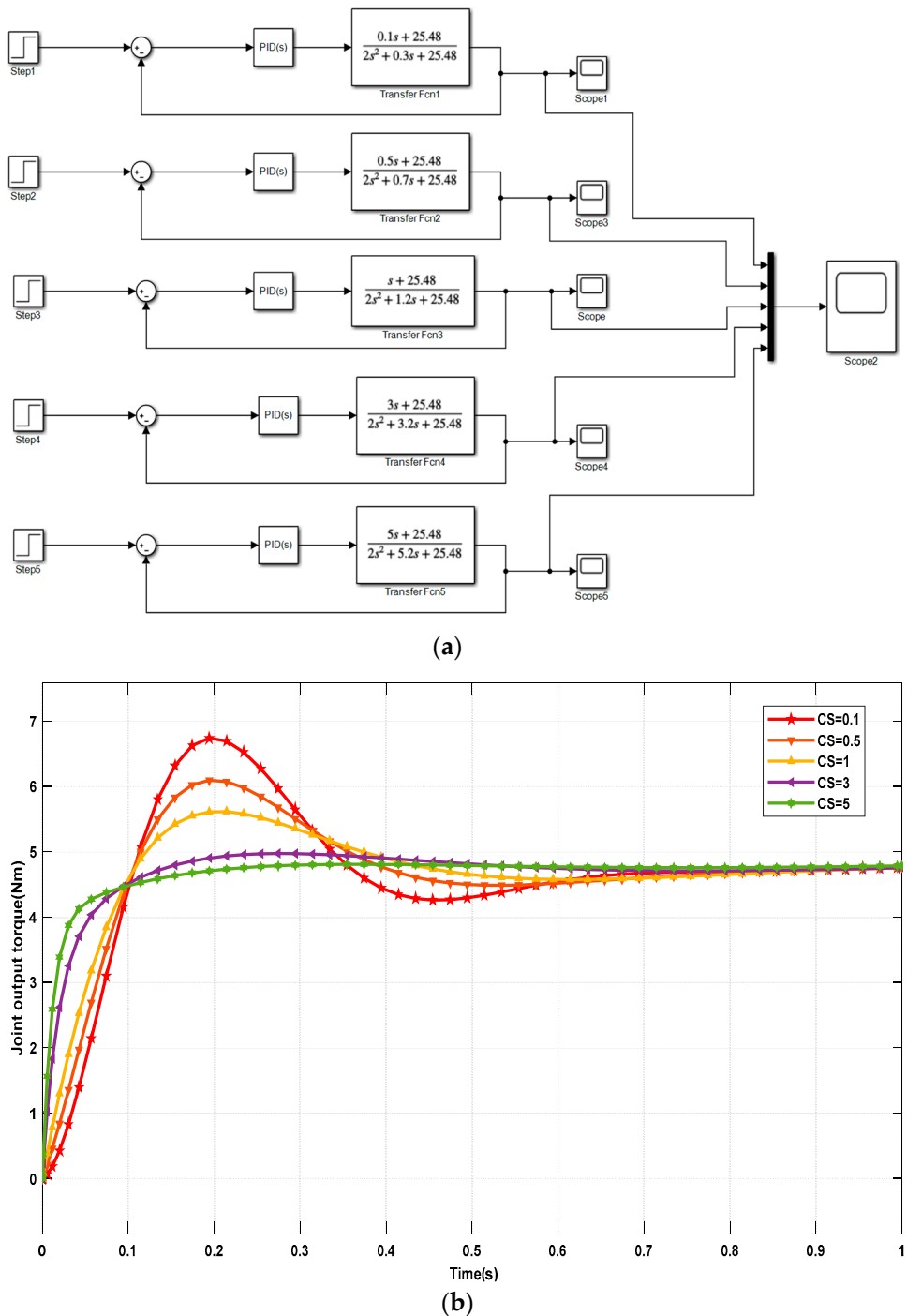

**(a)**

**(b)**

**Figure 20.** Output torque variation curve for different damping coefficients: (**a**) Simulink control system; (**b**) output torque variation curve.

Stability analysis of the open- and closed-loop transfers of the SEDA was conducted through Bode and Nyquist plots. This analysis allows us to establish the relationship between spring stiffness coefficient and driver stability. It also indicates the significant influence of the damping coefficient on SEDA's stability. Furthermore, it becomes evident that the closed-loop system achieves more precise control compared to the open-loop system, enabling the determination of PID control parameters for SEDA that meet the specified requirements.

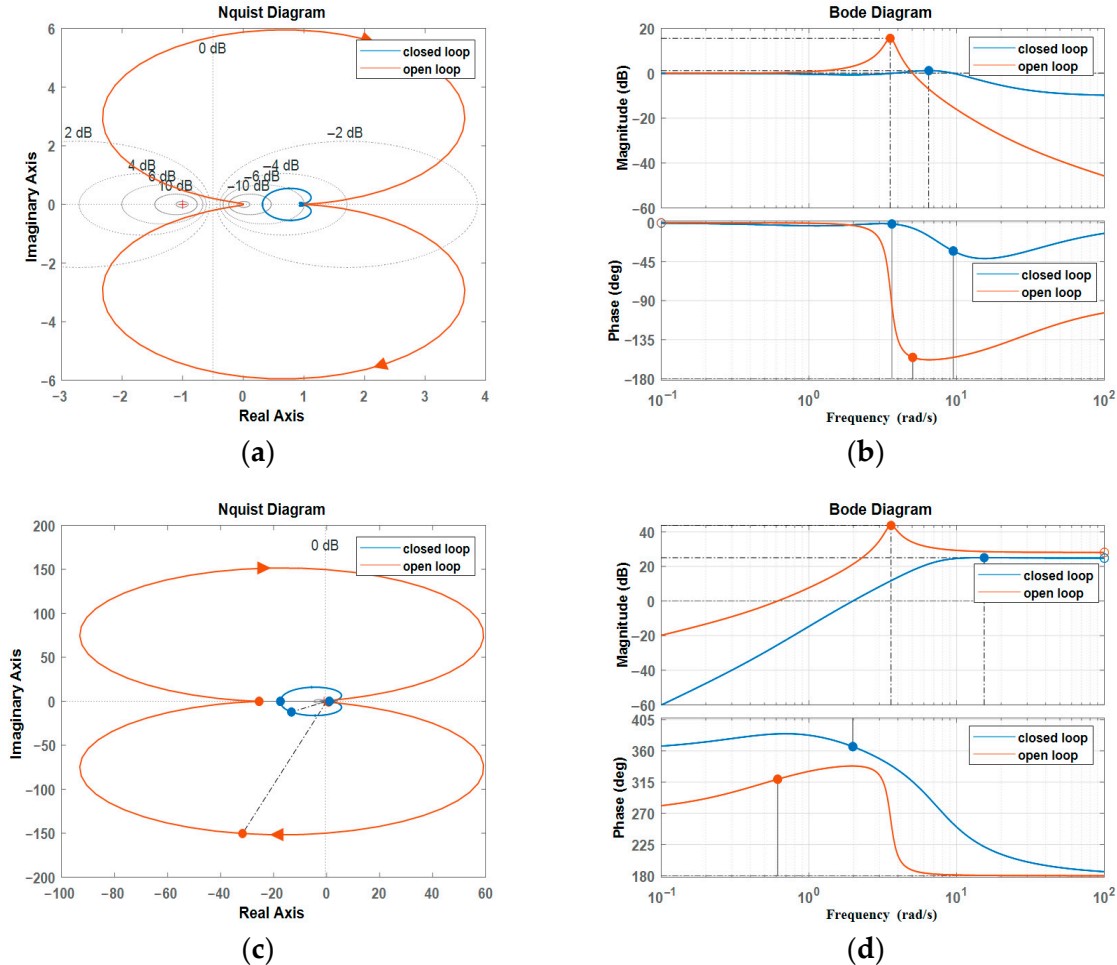

**Figure 21.** Open-loop and closed-loop system stability comparison analysis: (**a**) output bandwidth Nyquist diagram comparison; (**b**) output bandwidth Bode diagram comparison; (**c**) output impedance Nyquist diagram comparison; (**d**) output impedance Bode diagram comparison.

## 5. SEDA Fuzzy PID Controller Design and Simulink Simulation

In the process of assisting walking with lower limb exoskeleton rehabilitation robots, the environment is not static. Maintaining constant PID values may not yield optimal feedback effects. The SEDA, as a flexible driving mechanism, possesses certain shock resistance and damping effects. However, for the SEDA to interact more effectively with the external environment, the incorporation of compliant control strategies becomes essential. Fuzzy control is a control method based on fuzzy logic, primarily employed to handle systems that are difficult to model accurately or involve uncertainty factors [18]. Fuzzy control has a wide working range and a wide range of adaptability, especially suitable for the control of nonlinear systems. It does not depend on the mathematical model of the object, the complex object that cannot be modeled, and can also use knowledge from human experience to design a fuzzy controller to complete the control task, whereas traditional control methods need to know the mathematical model of the controlled object in order to design the controller. The fuzzy controller has an intrinsic parallel processing mechanism, exhibits strong robustness, is insensitive to changes in the characteristics of the controlled object, and the design parameters of the fuzzy controller are easy to select and adjust.

### 5.1. Fuzzy PID Controller Design

5.1.1. Domain and Membership Functions of the Fuzzy PID Controller

The inputs of the fuzzy PID controller are the error (*e*) and the rate of change of error (*ec*). The error is obtained by comparing the feedback value with the control setpoint.

The outputs are the three tuning parameters of the PID controller: proportional gain ($k_p$), integral gain ($k_i$), and derivative gain ($k_d$). The domains for the inputs $e$ and $ec$ in the fuzzy inference system are both [−6, 6], corresponding to linguistic values {Negative Big (NB), Negative Medium (NM), Negative Small (NS), Zero (ZO), Positive Small (PS), Positive Medium (PM), Positive Big (PB)}. Triangular membership functions (trimf) are employed for defining the membership degrees. For the outputs $k_p$, $k_i$, and $k_d$, the domain is also [−3, 3], with linguistic values {Negative Big (NB), Negative Medium (NM), Negative Small (NS), Zero (ZO), Positive Small (PS), Positive Medium (PM), Positive Big (PB)}. Similarly, triangular membership functions (trimf) are utilized for establishing the degrees of membership [19].

### 5.1.2. Fuzzy PID Controller Control Rules

The fuzzy control rules are the most crucial part of the fuzzy PID controller, determining the control accuracy and performance of the controller. To achieve optimal control results at different values of $e$ and $ec$, the self-adjustment rules for $k_p$, $k_i$, and $k_d$ are summarized as shown in Table 2.

**Table 2.** Corresponding values of $k_p$, $k_i$, $k_d$ in fuzzy PID control rules.

| $e$ | $ec$ | | | | | | |
| | NB | NM | NS | ZO | PS | PM | PB |
|---|---|---|---|---|---|---|---|
| NB | PB/NB/PS | PB/NB/PS | PM/NB/ZO | PM/NM/ZO | PS/NM/ZO | PS/ZO/PB | ZO/ZO/PB |
| NM | PB/NB/PS | PB/NB/NS | PM/NM/NS | PM/NM/NS | PS/NS/ZO | ZO/ZO/NS | ZO/ZO/PB |
| NS | PM/NM/NB | PM/NM/NB | PS/NM/NS | PS/NS/NS | ZO/ZO/ZO | NS/PS/PS | NM/PS/PM |
| ZO | PM/NM/NB | PS/NS/NM | PS/NS/NM | ZO/ZO/NS | NS/PS/ZO | NM/PS/PS | NM/PM/PM |
| PS | PS/NS/NB | PS/NS/NM | ZO/ZO/NS | NS/PS/NS | NS/PS/ZO | NM/PM/PS | NM/PM/PS |
| PM | ZO/ZO/NM | ZO/ZO/NS | NS/PS/NS | NM/PM/NS | NM/PM/ZO | NM/PB/PS | NB/PB/PS |
| PB | ZO/ZO/PS | NS/ZO/ZO | NS/PS/ZO | NM/PM/ZO | NM/PB/ZO | NM/PB/PB | NB/PB/PB |

Based on the aforementioned table, a total of 49 fuzzy control rules can be derived as follows:

If ($e$ is NB) and ($ec$ is NB), then ($k_p$ is PB)($k_i$ is NB)($k_d$ is PS);
If ($e$ is NB) and ($ec$ is NM), then ($k_p$ is PB)($k_i$ is NB)($k_d$ is NS);
If ($e$ is NB)and ($ec$ is NS), then ($k_p$ is PM)($k_i$ is NM)($k_d$ is NB);
If ($e$ is NB)and ($ec$ is ZO), then ($k_p$ is PM)($k_i$ is NM)($k_d$ is NB);
. . . . . .

The method of conjunction (AND) is taken as "min", and the method of disjunction (OR) is taken as "max". The implication method for inference is "min". The aggregation method is "max", and the defuzzification method is centroid (weighted average). By using a surface observation window, the output surfaces of $k_p$, $k_i$, and $k_d$ on the domain can be visualized, as shown in Figure 22.

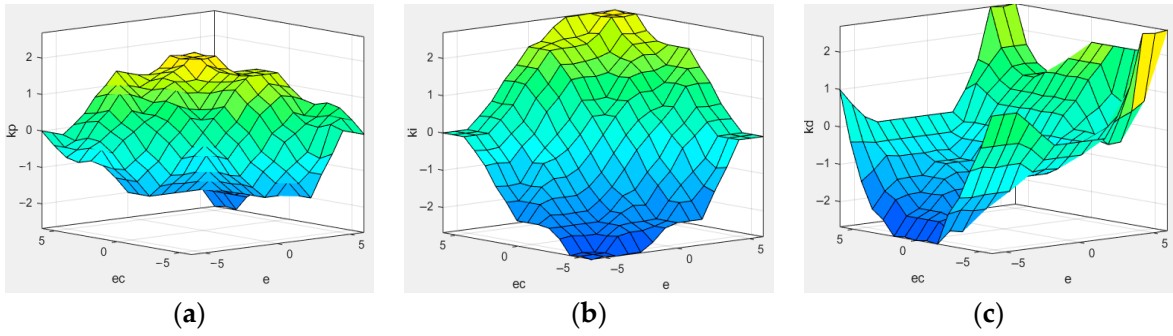

(**a**)  (**b**)  (**c**)

**Figure 22.** Output surface plots of $k_p$, $k_i$, $k_d$ on their respective domains: (**a**) $k_p$ output surface; (**b**) $k_i$ output surface; (**c**) $k_d$ output surface.

*5.2. Fuzzy PID Controller Modeling and Simulation*

5.2.1. Parameter Optimization of PSO Algorithm

The fuzzy factor $k_e$ error for the input value and the feedback value of the maximum error range for its domain range are equivalent in the same way the $k_{ec}$ error rate of change is equivalent to the displacement of the acceleration; the defuzzification factor $k_{p1}$ domain range for the output of $k_p$ is equivalent to the product of the range of changes in $k_{i1}$ and $k_{d1}$. In the same way, these factors can be iteratively analyzed by intelligent algorithms such as PSO (a particle swarm algorithm) to obtain the optimal fitness value of the best value of the five parameters.

Particle swarm optimization (PSO), a population-based stochastic optimization technique, was proposed by Eberhart and Kennedy in 1995 [20]. Particle swarm algorithms mimic the swarming behavior of insects, herds of animals, flocks of birds, schools of fish, etc. These groups search for food in a cooperative manner, and each member of the group constantly changes its search pattern by learning from its own experience and the experience of the other members [21]. The PSO first initializes a group of particles $N$, and then finds the optimal solution through an iterative process, and in each iteration, the particles update the global optimal positional extremes (OPEs) and the global optimal positional extremes (GPEs) by tracking the individual OPEs and the global optimal positional extremes to update their speed and position; its optimization formula is given in Equation (21):

$$V_{i+1} = \omega V_i + c_1 rand(t)(pBest[i] - X_i) + \\ c_2 rand(t)(prand(t)(pBest[g] - X_i) \tag{21}$$

where $X_{i+1} = X_i + V_{i+1}$; $V_i$ is the $i$-th particle's evolutionary speed; $X_i$ is the position of the $i$-th particle; $pBest[i]$ is the best position experienced by the $i$-th particle; $g$ indicates the position of the best particle in the population; $\omega$ is the inertia weight (adjusting its value can change the search range and the search speed) and usually set as 0.4–1.2; $c_1$ and $c_2$ are the learning factors, which are non-negative and usually set as $c_1 = c_2 = 2$; $rand(t)$ is the stochastic function to produce a random value of [0 1]. Each particle has a fitness value determined by a customized objective function, and each particle stores the current optimal value searched by itself and the optimal value in the current population, and dynamically adjusts this information as experience.

The fuzzy PID controller parameter optimization objective function, i.e., the fitness function, selects the integral performance index of the system, as shown in Equation (22):

$$J = \int_0^{t_s} t|e(t)|dt \tag{22}$$

where $t_s$ is the simulation time, which is the input and output error of the transfer function, and the ITAE criterion is used for optimization, which gives less consideration to the initial error of the system and mainly restricts the error that occurs at the later stage of the transition process. The optimized system is generally characterized by fast, smooth operation and a small overshoot.

The block diagram of the fuzzy PID controller optimized by the PSO algorithm is shown in Figure 23.

The parameters of the PSO algorithm are set as follows: the number of populations is 50; the dimension of each particle is D = 5; the initial values of the defuzzification factors *ke* and *kec* and defuzzification factors $k_{p1}$, $k_{i1}$, and $k_{d1}$ are set to [1,1,1,1,1,1]; the maximum number of iterations is 100; $c_1 = c_2 = 2$; $\omega = 0.95$; the velocity range of the particles is [−500, 500]; the particle position range is [−4, 4]; and finally, the PSO particle swarm algorithm is used to optimize and obtain the fuzzification factors *ke* = 0.8 and *kec* = 0.2 and defuzzification factors $k_{p1} = 0.5$, $k_{i1} = 8$, and $k_{d1} = −0.1$.

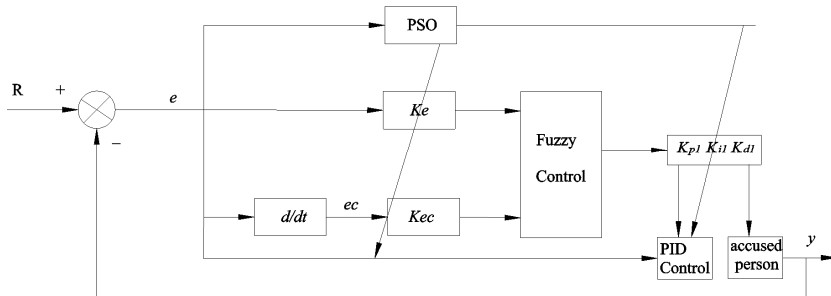

**Figure 23.** Block diagram of PSO algorithm parameter optimization system.

### 5.2.2. Fuzzy PID Controller Modeling

Creating System Simulation Model of Fuzzy PID Controller in MATLAB/Simulink Software version R2018a. In the MATLAB/Simulink software, a system simulation model is established, which mainly consists of a fuzzy PID controller, a disturbance signal, a controlled system (plant), and a control setpoint, as described in [22]. Combining the closed-loop system transfer function and the PID parameters established earlier, a comparison is made between no PID control, traditional PID control, and fuzzy PID control by constructing a Simulink diagram, as shown in Figure 24. The fuzzy factors, defuzzification factors, and initial values are the same as above.

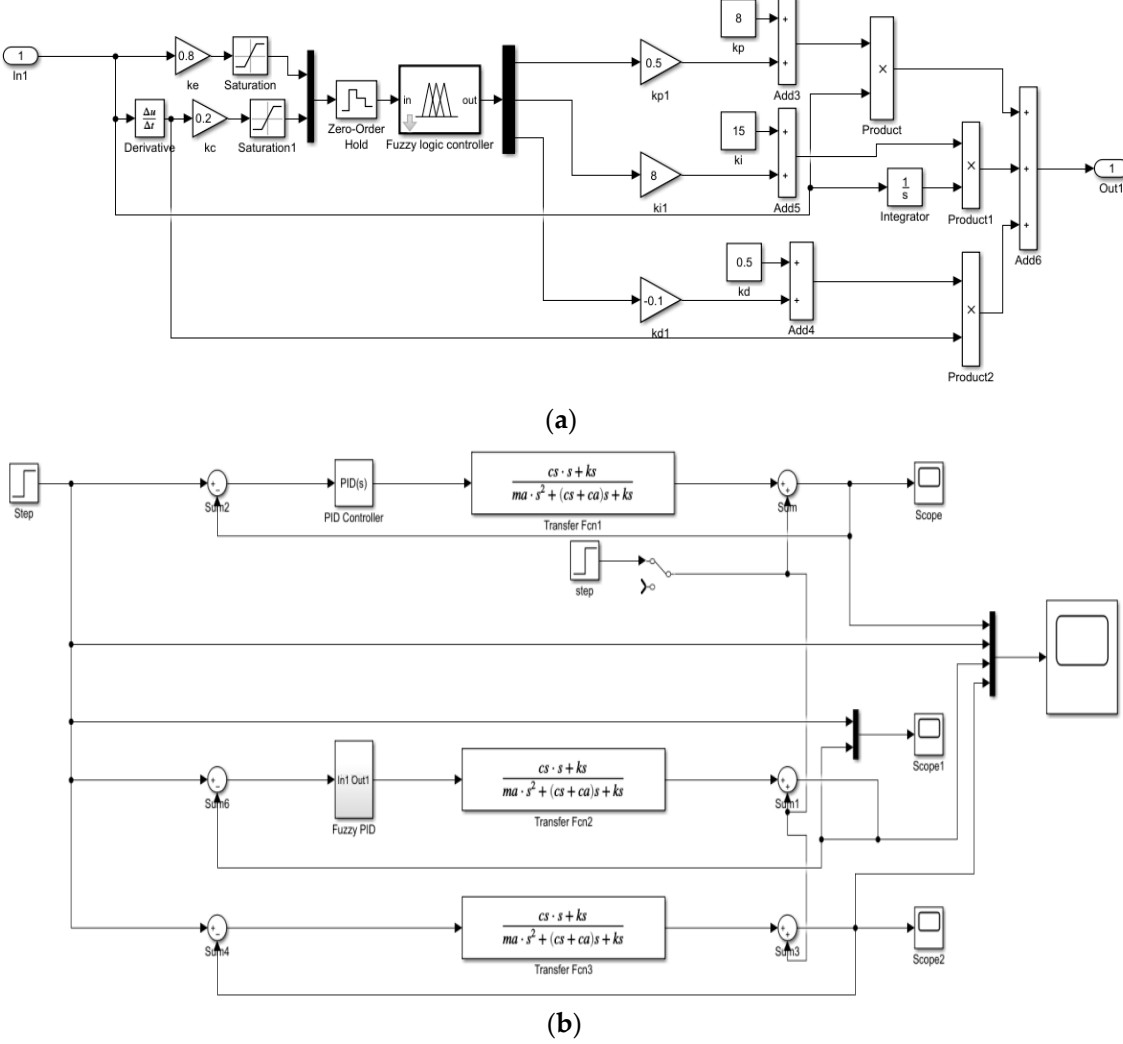

**Figure 24.** Simulink diagram of fuzzy PID control: (**a**) fuzzy PID controller subsystem diagram; (**b**) comparison diagram of 3 types of controllers.

### 5.2.3. Simulation Analysis of Fuzzy PID Controller

When a step signal of 10 Nm is input, the torque output of the joint shown in Figure 25a represents the simulation without disturbance, and Figure 25b represents the simulation with added disturbance.

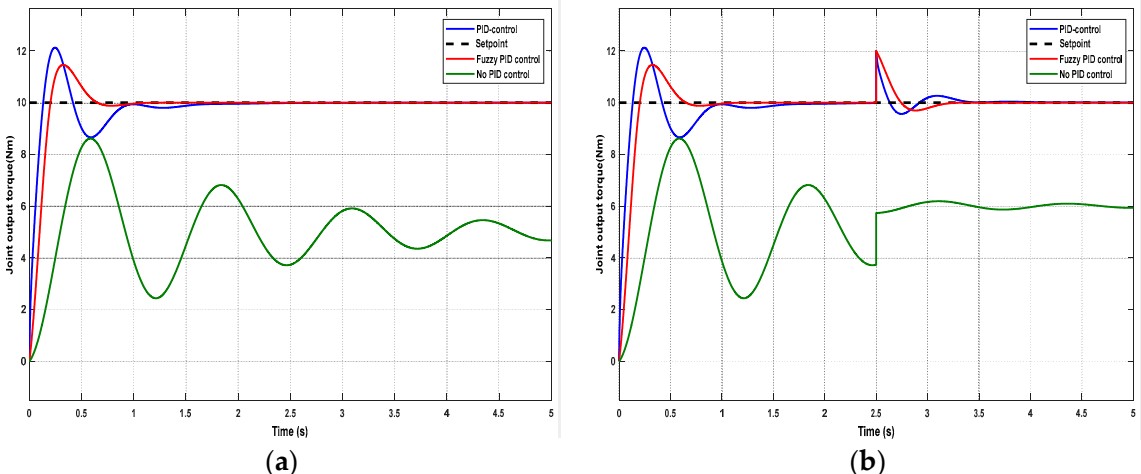

**(a)**            **(b)**

**Figure 25.** Simulink simulation results: (**a**) no disturbance; (**b**) with disturbance.

From the simulation curve in Figure 25a, it can be seen that when no PID control is used, the oscillation is obvious, the amount of overshoot is 47.3%, and the regulation time is about 5 s. When the traditional PID control is used, the oscillation is more obvious, the amount of overshoot is 21.2%, and the regulation time is about 0.96 s. When the fuzzy PID control is used, the system's oscillation is obviously weakened, with the amount of overshoot being 14.6%, and the regulation time is 0.66 s. The comparison of the simulation results of the three control methods are shown in the simulation results comparison table in Table 3.

**Table 3.** Comparison table of simulation results.

| Control Method | Overshoot | Stabilization Time |
| --- | --- | --- |
| No PID control | 47.3% | 5 s |
| PID control | 21.2% | 0.96 s |
| Fuzzy PID control | 14.6% | 0.66 s |

In the human body's actual walking process, the SEDA system will be disturbed by a variety of external factors. In order to verify the robustness of the system, the disturbance is added when the system is running for 2.5 s, and Figure 25b shows the simulation results. From the figure, it can be seen that when the disturbance occurs, the fuzzy PID controller has the fastest response time and the shortest regulation time, the PID controller is the second, and the no PID control is the slowest; furthermore, the oscillation amplitude under the fuzzy PID control is the smallest, the oscillation amplitude under the ordinary PID control is the second, and the oscillation amplitude under the no PID control is the largest.

### 6. Discussion

This study shows that the SEDA is better adapted to lower limb exoskeleton rehabilitation robots than conventional SEAs, because the SEDA adds a damping device on top of the elastic element, which constitutes a spring–damping closed-loop system that is more stable and has better shock resistance. The SEDA as an important drive system for lower limb exoskeleton rehabilitation robots, and can also be characterized by high response frequency, accurate control accuracy, short adjustment time, good dynamic performance, etc. The SEDA drive system is a kind of complex high-order nonlinear system with uncer-

tainty which is time-varying and susceptible to external disturbances, and relying on the conventional PID control cannot meet the control requirements. For the conventional PID control, although relatively simple and easy to operate, its relevant parameters are set in advance cannot be changed, so it cannot adapt to parameter changes or interference from more control systems, such as the use of fuzzy control. Although it can overcome some of the shortcomings of the PID algorithm, the steady-state accuracy is low, the dynamic performance is poor, and the control effect is very undesirable, among other shortcomings [23]. Combining the fuzzy control algorithm with the conventional PID control, the fuzzy PID control algorithm is obtained. Such a combination can not only retain the strengths of both, but also make up for the shortcomings of both, and ultimately achieve excellent control results.

Finally, it should be noted that in the simulation in this study, the assumed load force limit is 1000 N; thus, there are some limitations in the spring parameters and stiffness coefficients, the mass of the motor and the screw, the damping coefficients of the motor, and the damping coefficients of the damper in the selection, which results in the basic stiffness and damping of the fuzzy PID controller needing to be adjusted according to the symptomatic degree of the residual muscle strength of the patients with dyskinesia or the recovery of the muscle strength of the lower limb degree to be adjusted. In addition, there are many methods of tuning the PID controller parameters, such as theoretical calculation tuning methods and engineering tuning methods. Secondly, the development of the fuzzy control rule table is a complex process and does not have uniqueness, which may affect the generalizability of the research results. Therefore, future work will further improve the robot-assisted patient training evaluation system on the basis of this study to provide a basis for optimizing the system control parameters and rehabilitation training strategies.

## 7. Conclusions

Lower limb exoskeleton rehabilitation robots play a crucial role in promoting rehabilitation, enhancing muscle strength, improving gait, and enhancing quality of life. In this paper, aiming at the insufficient flexibility of traditional SEAs and the issues of wearability and safety they bring, a damping element is added to the base design to create a series elastic actuator with damping (SEDA). Finite element strength analysis was conducted under radial or axial loads to verify the feasibility of the designed structure. The analysis results indicate that the SEDA's structure and material selection meet the usage requirements.

By performing dynamic modeling of the SEDA, utilizing Bode and Nyquist plots, we conducted a comparative and analytical study of the open-loop and closed-loop frequency domains. We explored the effects of spring stiffness and damper damping coefficient on the system's output bandwidth, output impedance, and shock resistance. The research results demonstrate that as the stiffness coefficient increases, the response speed becomes faster, the rise time decreases, and the tracking performance improves. However, excessively high stiffness values can lead to a certain degree of overshoot and compromise system stability. As the damping coefficient increases, the system's response speed improves, stability enhances, and torque tracking performance becomes better. Yet, excessive damping can consume more energy. Ultimately, we selected a stiffness coefficient of $k_s$ = 25.48 N/mm for the elastic element and a damping coefficient of $c_s$ = 1 Ns/mm for the damper element.

Finally, according to the dynamics model of the SEDA system, a control scheme is proposed after simplified analysis. In order to make the flexible joints interact well with the external environment and to ensure the safety performance of the joints, a fuzzy controller is introduced, and the step response of the system is analyzed using MATLAB/Simulink simulation. It is concluded that, after adopting the fuzzy PID control, the control system has the following advantages: the fuzzy PID control method is used and the control system has the advantages of a higher stability, smaller steady state error, stronger resistance to external load interference, etc. Although the response time is a little slower in terms of dynamic performance, at the same time, the fuzzy PID control method reduces the

amount of overshooting, reduces the overshooting time, and has the ability to adapt to the environmental changes, as well as having a better control effect.

**Author Contributions:** Conceptualization, C.Z. and Z.L.; methodology, C.Z.; software, C.Z., Z.L. and L.Z.; validation, C.Z. and Y.W.; writing—original draft preparation, C.Z.; writing—review and editing, C.Z.; supervision, Z.L.; project administration, L.Z. All authors have read and agreed to the published version of the manuscript.

**Funding:** This research is supported by the Special Fund for Bagui Scholars of the Guangxi Zhuang Autonomous Region No. 2019A08, the Guangxi Higher Education Undergraduate Teaching Reform Project of 2023 No. 2023JGA249, the 2019 Research Initiation Program for Introducing High-level Talents to Beibu Gulf University No. 2019KYQD03, Guangxi Key Laboratory of Ocean Engineering Equipment And Technology, the Key Laboratory of Beibu Gulf Offshore Engineering Equipment, and Technology (Beibu Gulf University), Education Department of Guangxi Zhuang Autonomous Region. And Technology (Beibu Gulf University), Education Department of Guangxi Zhuang Autonomous Region, Qinzhou 535011, China.

**Data Availability Statement:** Data are contained within the article.

**Conflicts of Interest:** The authors declare no conflicts of interest.

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
