# Peer review of "Design and Research of Series Actuator Structure and Control System Based on Lower Limb Exoskeleton Rehabilitation Robot"

_actuators, doi:10.3390/act13010020_

Round 1

Reviewer 1 Report

Comments and Suggestions for Authors

The authors developed a Series Elastic-Damping Actuator (SEDA) to address limitations in traditional elastic actuators for lower limb exoskeleton rehabilitation robots. Through finite element static analysis, dynamic modeling, and frequency domain comparisons, they validated the feasibility of the design. System stability, impact of the damping coefficient, and component parameters were assessed. The study incorporated a fuzzy controller in the PID control system, simulating joint torque output under various scenarios. MATLAB/Simulink analysis confirmed the effectiveness of the design structure and control method in enhancing flexible joint output. However, some critical concerns need to be addressed before publication. 

1. The key findings should be included at the end of the Abstract. The works on SEDAs should be referred and then the authors should tell the novel contributions of their work as compared to existing ones. The motivation to design a new SEDA is unclear in the manuscript.

2. How did the authors determine the feasibility of the design and the selection of materials through finite element static analysis?? There should be more information on the number of elements, nodes, element type, and size considered during finite element analysis.

3. Section 4 should come earlier than Section 3 as after deciding the static strength, one can check for the dynamic analysis. Line 170,175, the references need to be cited. The notations such as m_a, c_a, c_s, and k_s (in running text) should be checked for correctness. How the parameters in Table 1 are selected?

4. What motivated the choice of incorporating a fuzzy controller into the PID control system? How the fuzzy and de-fuzzy in lines 516-518 are selected? The authors are further suggested to elaborate on the simulations conducted to assess joint torque output under external disturbance and no external disturbance scenarios. What about addressing model uncertainties?

5. How do the results confirm the effectiveness of the design structure and control method in enhancing flexible joint output? How does the present work compare to existing approaches or similar studies in the field? Asking that, the results should be strengthened.

6. There are many robust control algorithms for the lower-limb exoskeletons in the literature. The authors are suggested to elaborate on the reasoning behind incorporating a fuzzy controller in the PID system.

7. The authors should present a discussion section before the conclusion and summarize the contributions in the conclusion. Were there any limitations or assumptions made in the simulation that may affect the generalizability of the findings?

Reviewer 2 Report

Comments and Suggestions for Authors

In order to better achieve human-machine interaction and the goal of flexible driving, and in response to the shortcomings of traditional elastic actuators, a series elastic damping actuator (SEDA) was designed. This research has certain academic value, but still requires a lot of improvement and refinement. In addition, there are some innovative points in this research, but there are still some shortcomings in the experimental design.

1. When verifying the impact resistance of SEDA (3.2.3), the authors mentioned: "During the walking process of lower limb exoskeleton rehabilitation robots, when the feet contact the ground, significant impact will be exerted on the robot's body, even causing damage to its parts. The ground exerts a significant impact on the robot's body, even causing damage to its parts. It is suggested to provide the GRF curve of the lower limb exoskeleton rehabilitation robot during walking, to make it more convincing."

2. When introducing the Bode plot and Nyquist plot in the article, it is recommended to summarize and analyze the content of the images, rather than just describing the data in the images.

3. In the introduction, the author mentions "comparing open-loop and closed-loop systems as well as damped and undamped systems to analyze the impact of spring constant and damping factor on system stability".
Damping, analysis of the effects of elastic modulus and damping coefficient on system stability. It is recommended here to control one variable, give a summary of the results, and suggest comparing open-loop damping systems with open-loop damping systems or open-loop damping systems with closed-loop damping systems to draw conclusions.

4. It was mentioned in the article that "by using the Optitrack optical 3D motion capture experiment, it was observed that significant torque is generated at the hip and knee joints during normal walking. However, a thorough reading of the entire paper did not show the torque at the hip and knee joints."
I suggest adding a comparative experiment of before and after optimization in terms of measured angles and torque.

5. The experimental design and method description are not detailed enough, and I suggest the authors provide more experimental details for readers to reproduce the research results.

Comments on the Quality of English Language

1. The sentence structure is not clear enough. I suggest the author simplify the sentence and avoid using overly complex vocabulary and sentence structures.
2. The transition between paragraphs is not smooth. I suggest the author use appropriate transition words and sentences to connect paragraphs and ideas.

Author Response

Comments 1: When verifying the impact resistance of SEDA (3.2.3), the authors mentioned: "During the walking process of lower limb exoskeleton rehabilitation robots, when the feet contact the ground, significant impact will be exerted on the robot's body, even causing damage to its parts. The ground exerts a significant impact on the robot's body, even causing damage to its parts. It is suggested to provide the GRF curve of the lower limb exoskeleton rehabilitation robot during walking, to make it more convincing."

Response 1: In Section 2, barefoot normal walking GRF force curves were collected, and in verifying the impact resistance of SEDA (3.2.3), the GRF curves of the lower limb exoskeleton rehabilitation robot during walking were simulated and compared to illustrate that, "Compared to the barefoot walking GRF graph, the impact peak is missing, the slope of the line segment in the initial phase has become flat, and the vertical impact rate is reduced, which is mainly due to the elastic and damping elements in the SEDA eliminating part of the impact force and cushioning the vibration."

Comments 2: When introducing the Bode plot and Nyquist plot in the article, it is recommended to summarize and analyze the content of the images, rather than just describing the data in the images.

Response 2: The Bode and Nyquist plots appearing in the article are analyzed and summarized, the stability of the open-loop and closed-loop system functions, the amplitude-following characteristics and phase-following characteristics of the system appearing in the low-frequency and high-frequency bands, as well as the amount of overshooting that occurs are analyzed.

Comments 3: In the introduction, the author mentions "comparing open-loop and closed-loop systems as well as damped and undamped systems to analyze the impact of spring constant and damping factor on system stability".Damping, analysis of the effects of elastic modulus and damping coefficient on system stability. It is recommended here to control one variable, give a summary of the results, and suggest comparing open-loop damping systems with open-loop damping systems or open-loop damping systems with closed-loop damping systems to draw conclusions.

Response 3:In Chapter 4.2, mainly using the elasticity coefficient as a single variable, the influence of the elasticity coefficient on the system stability of the system in the open-loop undamped system is analyzed, and the value of the elasticity coefficient is determined; in Chapter 4.3, mainly using the damping coefficient as a single variable, the open-loop undamped system and the open-loop damped system are compared, and the influence of the damping coefficient on the system stability of the system is analyzed. The influence of the damping coefficient is determined, and the value of the damping coefficient is determined; in chapter 4.4, the open-loop damped system is mainly compared with the closed-loop damped system, and it is finally concluded that the closed-loop damped system can realize more accurate control and obtain the PID control parameters of SEDA that meet the requirements.

Comments 4: It was mentioned in the article that "by using the Optitrack optical 3D motion capture experiment, it was observed that significant torque is generated at the hip and knee joints during normal walking. However, a thorough reading of the entire paper did not show the torque at the hip and knee joints."I suggest adding a comparative experiment of before and after optimization in terms of measured angles and torque.

Response 4:In Section 2, the angle change curves, torque change curves, and GRF curves of the hip, knee, and ankle joints captured by using Optitrack optical 3D motion capture experiments are supplemented. Since the angle change and torque change of the lower limb exoskeleton rehabilitation robot do not change much before and after the optimization, the optimized GRF curves are added to demonstrate the importance of the elastic and damping elements in SEDA.

Comments 5: The experimental design and method description are not detailed enough, and I suggest the authors provide more experimental details for readers to reproduce the research results.

Response 5:In Section 2, the experimental procedure was added: "The experiment was conducted using Quanser's Optitrack optical 3D motion capture system equipped with six FLEX 3 infrared cameras to achieve spatial 3D localization, together with a 3D force measuring table and Motive data analysis software. A three-dimensional space with a length of 5 meters, a width of 4.4 meters and a height of 2.6 meters was installed indoors, and one healthy male subject was selected to participate in the lower limb joint motion capture experiment (age 25 years old, body weight of 75Kg, height of 175cm), taking walking as the basic motion mode, taking the horizontal road surface as the basic constraints, and combining the characteristics of the muscle activity groups in the human body's walking with the passive infrared optics reflective principle, the lower limbs in the human body were positioned in a non-linear, non-contrasting, non-directional, and non-directional direction. Lower limbs in accordance with the non-linear, asymmetric way to paste 17 Marker points, barefoot walking, and three-dimensional force table synchronization of the human lower limb motion data acquisition, sampling frequency of 100Hz. so that you can get the human body's various joints of the motion parameters and GRF (Ground Support Return Force) curve."

4. Response to Comments on the Quality of English Language

Point 1: The sentence structure is not clear enough. I suggest the author simplify the sentence and avoid using overly complex vocabulary and sentence structures.

Response 1:I've simplified some of the statements in the article.

Point 2: The transition between paragraphs is not smooth. I suggest the author use appropriate transition words and sentences to connect paragraphs and ideas.

Response 2:I added some transition words and sentences between paragraphs to make sure the transitions were smooth.

Once again, I would like to express my sincere gratitude to the editors and reviewers for their enthusiastic work, and I hope that the revisions will be recognized, and thank you again for your comments and suggestions.

Round 2

Reviewer 1 Report

Comments and Suggestions for Authors

Although the authors have tried to address some of the concerns, a few critical issues still need to be dealt with. The authors should revise the manuscript carefully given the following comments.

1. The key findings numerical should be included at the end of the Abstract. In the Introduction, more relevant works on SEDAs should be included to improve the knowledge base and then the authors should tell the novel contributions of their work as compared to existing ones. 

2. It is still unclear how the authors have selected fuzzy and defuzzy parameters. Have you used PSO as you mentioned in the rebuttal? If yes, where are the details, and if no, how?

3. How does the present work compare to existing approaches or similar studies in the field? Asking that, the results should be strengthened by comparing quantitatively with similar works.

4. There are many robust control algorithms for the lower-limb exoskeletons in the literature. The authors are suggested to elaborate on the reasoning behind incorporating a fuzzy controller in the PID system. The authors should focus on the advantages of fuzzy application rather than ambiguous claims about adaptive and robust control schemes.

5. There are two points made by the authors that are somehow incorrect. First, the adaptive and robust control schemes are part of modern control rather than traditional control theory. Second, the authors have given insufficient information on the limitations of robust control by saying that 'but the effect of this control method is very limited', how?

Author Response

Comments 1: The key findings numerical should be included at the end of the Abstract. In the Introduction, more relevant works on SEDAs should be included to improve the knowledge base and then the authors should tell the novel contributions of their work as compared to existing ones. 

Response 1: In the existing research work, some scholars use high stiffness and high impedance rigid structures as the joints of exoskeleton robots, which have high localization accuracy but poor flexibility. In order for the robot to cope with the dynamic changes of the working environment and the uncertainty of human-robot interaction, the joints of the robot must have a certain degree of suppleness and safe human-robot interaction, and some scholars have isolated the shock load between the motor end and the load end by adding elastic elements to the robot joints, which constitutes a typical series elastic actuator structure, so in the introduction, the research on SEA is mainly introduced Situation. Compared with the existing work, my main contributions, which are added and summarized in the introduction, can be roughly divided into the following points:

(1) Most researchers simplify SEA as pure stiffness link or damping link when simplifying the kinetic model of elastic element, and the kinetic model is imperfect. Therefore, I improve the SEA and propose a structure combining the spring and damper, i.e., SEDA, to establish its kinetic model, obtain the open-loop and closed-loop system transfer functions driven by the force source through the Laplace transform, and then analyze its stability through the stability criterion of the Nyquist diagram, and obtain the force output bandwidth and output impedance through the Bode diagram, which proves the importance of adding the damping The importance of adding damping elements is demonstrated.

(2) In the dynamic model analysis, many scholars regard the motor as an ideal speed output source or position output source, ignoring the inertia characteristics of the motor itself, resulting in a large error, I regard the motor as an ideal force output source, due to the motor output force in the range of the output capacity is directly proportional to the excitation current (voltage), and for the SEDA such as a mechanical system, the time of the change of the excitation current is almost negligible, so simplifying the motor as a kind of force output source is relatively reasonable.

(3) In the drive control method, most did not consider the system itself exists in a variety of external disturbances, while the perturbation problem is unavoidable in the practical application, so the fuzzy PID control method is proposed to simplify the design of the controller and improve the anti-interference ability, and simulation analysis is carried out on the MATLAB/Simulink platform, which verifies the correctness of the design of the fuzzy PID controller.

Comments 2: It is still unclear how the authors have selected fuzzy and defuzzy parameters. Have you used PSO as you mentioned in the rebuttal? If yes, where are the details, and if no, how?

Response 2: I've added a small section 5.2.1 to Chapter 5.2 that supplements the PSO particle swarm algorithm parameter optimization.

Comments 3: How does the present work compare to existing approaches or similar studies in the field? Asking that, the results should be strengthened by comparing quantitatively with similar works.

Response 3:The main improvements of my current work compared to existing research in the field are as follows:

(1) Better dynamics modeling. In Section 4.1, the SEDA design schematic 13 is an improvement on the SEA design schematic 12 and yields a more refined dynamics model.

(2) Better stability. The current research only simply adds elastic elements to the rigid structure, and in Chapter 4.2, mainly using the elasticity coefficient as a single variable, the influence of the elasticity coefficient on the system stability of the system in the open-loop undamped system is analyzed, and the value of the elasticity coefficient is determined; in Chapter 4.3, mainly using the damping coefficient as a single variable, a comparison between the open-loop undamped system and the open-loop damped system is carried out, and the output bandwidth, output impedance, and shock resistance of the driver in the open-loop undamped system is analyzed. In the open-loop damped system, the output bandwidth, output impedance, and shock resistance of the driver are obtained as (a), (b), and (c) Bode plots in Fig. 18, and from this, it is concluded that the system has good amplitude-following characteristics and phase-following characteristics at both low frequency bands, and the phase curves do not have intersection points with -180°, and the system is well stabilized, and the resonant peaks are getting smaller and smaller as the damping coefficients are increased. smaller and the stability of the system is getting better and better. In section 4.4.1, the output torque of the SEDA closed-loop system is simulated, and the damping coefficient of the system is obtained by the time for the system to reach the steady state.

(3) Better impact resistance. In Section 2, barefoot normal walking GRF force curves were collected, and in verifying the impact resistance of SEDA (4.2.3), the GRF curves of the lower limb exoskeleton rehabilitation robot when walking were simulated and compared to illustrate that, "Compared to the barefoot walking GRF graphs, the peaks of the impacts are missing, and the slopes of the line segments in the initial phase have become flat, and the vertical impact rate is reduced, which is mainly due to the elastic and damping elements in the SEDA eliminating part of the impact force and cushioning the vibration."

(4) Control methods. There are many modern control methods, the article mainly in chapter 5, the simulation of no PID control, PID control and fuzzy PID in the amount of overshooting and stabilization time to compare and analyze, and concluded that the fuzzy PID controller has the smallest oscillation amplitude, the fastest regulation reaction time and the shortest regulation time.

Comments 4: There are many robust control algorithms for the lower-limb exoskeletons in the literature. The authors are suggested to elaborate on the reasoning behind incorporating a fuzzy controller in the PID system. The authors should focus on the advantages of fuzzy application rather than ambiguous claims about adaptive and robust control schemes.

Response 4:I restate the reasons for including a fuzzy controller in a PID system in Chapter 6:

SEDA as an important drive system for lower limb exoskeleton rehabilitation robots, should also be characterized by high response frequency, accurate control accuracy, short adjustment time, good dynamic performance, etc. The SEDA drive system is a kind of complex high-order nonlinear system with uncertainty, time-varying and sus-ceptible to external disturbances, and relying on the conventional PID control can not meet the control requirements. For the conventional PID control, although relatively simple and easy to operate, but the relevant parameters are set in advance can not be changed, so it can not adapt to the parameter changes, interference with more control systems, such as the use of fuzzy control, although it can overcome some of the short-comings of the PID algorithm, but there is still a steady state accuracy is low, the dy-namic performance of the poor, the control effect is very undesirable and other short-comings. Combining the fuzzy control algorithm with the conventional PID control, the fuzzy PID control algorithm is obtained. Such a combination can not only retain the strengths of both, but also make up for the shortcomings of both, and ultimately achieve excellent control results.

Comments 5: There are two points made by the authors that are somehow incorrect. First, the adaptive and robust control schemes are part of modern control rather than traditional control theory. Second, the authors have given insufficient information on the limitations of robust control by saying that 'but the effect of this control method is very limited', how?

Response 5:Thank you for the reviewer's criticism and correction, it is my misperception of adaptive control and robust control scheme, which is a modern control theory, and I have deleted the misdescription in the article. Adaptive control and robust control are both modern and excellent control methods that can cope with the situation when the mathematical model cannot accurately represent the actual system, it is the lack of my knowledge, if adaptive control or robust control methods are applied to SEDA, it should also achieve good control results. This will be one of my next study and research, trying to pursue a better control scheme in modern control theories such as adaptive control, robust control, neural network, fuzzy PID control and so on.

Once again, I would like to express my sincere gratitude to the editors and reviewers for their enthusiastic work, and I hope that the revisions will be recognized, and thank you again for your comments and suggestions.

Reviewer 2 Report

Comments and Suggestions for Authors

Thank the author for their hard work. The revised version has been carefully improved and the manuscript is very helpful to other researchers.

Author Response

I would like to thank the reviewing experts for recognizing my manuscript, and I will continue to work hard to do a good job in my research, and I wish you good health and success in your work.

Round 3

Reviewer 1 Report

Comments and Suggestions for Authors

The authors have addressed all major concerns raised by the reviwer. However, it would be nice to include some numerical findings of the present work at the end of the abstract. Also, whatever novelty the authors are presenting as compared to other works, all such existing works need to be cited. 

Author Response

Comments : The authors have addressed all major concerns raised by the reviwer. However, it would be nice to include some numerical findings of the present work at the end of the abstract. Also, whatever novelty the authors are presenting as compared to other works, all such existing works need to be cited.  

Response : (1)I have included some of the results of this study such as the selection of the elasticity coefficient and damping coefficient, the selection of the fuzzy and de-fuzzy factors after PSO optimization, and the fuzzy PID control results obtained through MATLAB/Simulink simulation, which have been marked in the abstract as follows:

By modeling the dynamics of the SEDA, using the Bode plot and Nyquist plot, the open-loop and closed-loop frequency domain comparisons and analyses were carried out, respectively, to verify the effect of damping coefficients on the stability of the system, and the stiffness coefficient ks=25.48N/mm as the elastic element and the damping coefficient cs=1Ns/mm as the damping element were selected. A particle swarm optimization (PSO)-based algorithm is proposed to in-troduce the fuzzy controller into the PID control system, and the fuzzy controller's fuzzy factor (ke,kec) and de-fuzzy factor (kp1,ki1,kd1) 5 parameters are taken as the object of the algorithm op-timization to get the optimal ke=0.8, kec=0.2,kp1=0.5, ki1=8, kd1=-0.1 fuzzy controller parameters. The joint torque output with and without external interference is simulated, and the simulation model is established in the MATLAB/Simulink environment, and the results show that: when fuzzy PID control is used, the overshooting amount of the system is 14.6%, and the regulation time is 0.66s, which has the following advantages: small overshooting amount, short rise time, fast response speed, short regulation time, good stability performance, and strong an-ti-interference ability. SEDA design structure and control method breaks through the traditional series elastic actuator (SEA) in the lack of flexibility and stability, which is very helpful to improve the output effect of flexible joints.

(2) Thanks to the corrections of the reviewers, I have added some references to the current study, which have been marked in the article, as well as renumbered the references of the paper.

Once again, I would like to express my sincere gratitude to the editors and reviewers for their enthusiastic work, and I hope that the revisions will be recognized, and thank you again for your comments and suggestions.
